# Solving Min-Max Optimization with Hidden Structure via Gradient Descent Ascent

**Lampros Flokas**[*]
Department of Computer Science
Columbia University
New York, NY 10025
`lamflokas@cs.columbia.edu`

**Emmanouil V. Vlatakis-Gkaragkounis**[*]
Department of Computer Science
Columbia University
New York, NY 10025
`emvlatakis@cs.columbia.edu`

**Georgios Piliouras**
Singapore University of Technology & Design
`georgios.piliouras@sutd.edu.sg`

## Abstract

Many recent AI architectures are inspired by zero-sum games, however, the behavior of their dynamics is still not well understood. Inspired by this, we study standard gradient descent ascent (GDA) dynamics in a specific class of non-convex non-concave zero-sum games, that we call hidden zero-sum games. In this class, players control the inputs of smooth but possibly non-linear functions whose outputs are being applied as inputs to a convex-concave game. Unlike general zero-sum games, these games have a well-defined notion of solution; outcomes that implement the von-Neumann equilibrium of the "hidden" convex-concave game. We provide conditions under which vanilla GDA provably converges not merely to local Nash, but the actual von-Neumann solution. If the hidden game lacks strict convexity properties, GDA may fail to converge to any equilibrium, however, by applying standard regularization techniques we can prove convergence to a von-Neumann solution of a slightly perturbed zero-sum game. Our convergence results are non-local despite working in the setting of non-convex non-concave games. Critically, under proper assumptions we combine the Center-Stable Manifold Theorem along with novel type of initialization dependent Lyapunov functions to prove that almost all initial conditions converge to the solution. Finally, we discuss diverse applications of our framework ranging from generative adversarial networks to evolutionary biology.

## 1 Introduction

Traditionally, our understanding of convex-concave games revolves around von Neumann's celebrated minimax theorem, which implies the existence of saddle point solutions with a uniquely defined value. These solutions are called *von Nemann solutions* and guarantee to each agent their corresponding value regardless of opponent play. Although many learning algorithms are known to be able to compute such saddle points [13], recently there has there has been a fervor of activity in proving stronger results such as faster regret minimization rates or analysis of the day-to-day behavior [46, 17, 7, 1, 66, 19, 2, 45, 5, 25, 70, 29, 6, 48, 30, 56].

This interest has been largely triggered by the impressive successes of AI architectures inspired by min-max games such as Generative Adversarial Networks (GANS) [26], adversarial training [40] and reinforcement learning self-play in games [63]. Critically, however, all these applications are based upon *non-convex non-concave games*, our understanding of which is still nascent. Nevertheless,

35th Conference on Neural Information Processing Systems (NeurIPS 2021).

some important early work in the area has focused on identifying new solution concepts that are widely applicable in general min-max games, such as (local/differential) Nash equilibrium [3, 41], local minmax [18], local minimax [31], (local/differential) Stackleberg equilibrium [24], local robust point [69]. The plethora of solutions concepts is perhaps suggestive that "solving" general min-max games unequivocally may be too ambitious a task. Attraction to spurious fixed points [18], cycles [65], robustly chaotic behavior [15, 16] and computational hardness issues [20] all suggest that general min-max games might inherently involve messy, unpredictable and complex behavior.

*Are there rich classes of non-convex non-concave games with an effectively unique game theoretic solution that is selected by standard optimization dynamics (e.g. gradient descent)?*

**Our class of games.** We will define a general class of min-max optimization problems, where each agent selects its own vectors of parameters which are then processed separately by smooth functions. Each agent receives their respective payoff after entering the outputs of the processed decision vectors as inputs to a standard convex-concave game. Formally, there exist functions $\mathbf{F} : \mathbb{R}^N \to X \subset \mathbb{R}^n$ and $\mathbf{G} : \mathbb{R}^M \to Y \subset \mathbb{R}^m$ and a continuous convex-concave function $L : X \times Y \to \mathbb{R}$, such that the min-max game is

$$\min_{\boldsymbol{\theta} \in \mathbb{R}^N} \max_{\boldsymbol{\phi} \in \mathbb{R}^M} L(\mathbf{F}(\boldsymbol{\theta}), \mathbf{G}(\boldsymbol{\phi})). \qquad \text{(Hidden Convex-Concave (HCC))}$$

We call this class of min-max problems Hidden Convex-Concave Games. It generalizes the recently defined hidden bilinear games of [65].

**Our solution concept.** Out of all the local Nash equilibria of HCC games, there exists a special subclass, the vectors $(\boldsymbol{\theta}^*, \boldsymbol{\phi}^*)$ that implement the von Neumann solution of the convex-concave game. This solution has a strong and intuitive game theoretic justification. Indeed, it is stable even if the agents could perform arbitrary deviations directly on the output spaces $X, Y$. These parameter combinations $(\boldsymbol{\theta}^*, \boldsymbol{\phi}^*)$ "solve" the "hidden" convex-concave $L$ and thus we call them *von Neumann solutions*. Naturally, HCCs will typically have numerous local saddle/Nash equilibria/fixed points that do not satisfy this property. Instead, they correspond to stationary points of the $\mathbf{F}, \mathbf{G}$ where their output is stuck, e.g., due to an unfortunate initialization. At these points the agents may be receiving payoffs which can be arbitrarily smaller/larger than the game theoretic value of game $L$. Fortunately, we show that Gradient Descent Ascent (GDA) strongly favors von Neumann solutions over generic fixed points.

**Our results.** In this work, we study the behavior of continuous GDA dynamics for the class of HCC games where each coordinate of $\mathbf{F}, \mathbf{G}$ is controlled by disjoint sets of variables. In a nutshell, we show that GDA trajectories stabilize around or converge to the corresponding von Neumann solutions of the hidden game. Despite restricting our attention to a subset of HCC games, our analysis has to overcome unique hurdles not shared by standard convex concave games.

*Challenges of HCC games.* In convex-concave games, deriving the stability of the von Neumann solutions relies on the Euclidean distance from the equilibrium being a Lyapunov function. In contrast, in HCC games where optimization happens in the parameter space of $\boldsymbol{\theta}, \boldsymbol{\phi}$, the non-linear nature of $\mathbf{F}, \mathbf{G}$ distorts the convex-concave landscape in the output space. Thus, the Euclidean distance will not be in general a Lyapunov function. Moreover, the existence of *any* Lyapunov function for the trajectories in the output space of $\mathbf{F}, \mathbf{G}$ does not translate to a well-defined function in the parameter space (unless $\mathbf{F}, \mathbf{G}$ are trivial, invertible maps). Worse yet, even if $L$ has a unique solution in the output space, this solution could be implemented by multiple equilibria in the parameter space and thus each of them can not be individually globally attracting. Clearly any transfer of stability or convergence properties from the output to the parameter space needs to be initialization dependent. It is worth mentioning that similar challenges like transfering results from the output to the input space was also faced in the simpler class of hidden bilinear games. However, [65] to sidestep this issue assume the restricitve requirement of $\mathbf{F}, \mathbf{G}$ to be invertible operators. Our results go beyond this simplified case requiring new proof techniques. Specifically, we show how to combine the powerful technologies of the the Center-Stable Manifold Theorem, typically used to argue convergence to equilibria in non-convex optimization settings [34, 52, 54, 53, 35], along with a novel Lyapunov function argument to prove that almost all initial conditions converge to the our game theoretic solution.

*Lyapunov Stability.* Our first step is to construct an initialization-dependent Lyapunov function that accounts for the curvature induced by the operators $\mathbf{F}$ and $\mathbf{G}$ (Lemma 2). Leveraging a potentially

infinite number of initialization-dependent Lyapunov functions in Theorem 5 we prove that under mild assumptions the outputs of $\mathbf{F}, \mathbf{G}$ stabilize around the von Neumann solution of $L$.

*Convergence.* Mirroring convex concave games, we require strict convexity or concavity of $L$ to provide convergence guarantees to von Neumann solutions (Theorem 6). Barring initializations where von Neumann solutions are not reachable due to the limitations imposed by $\mathbf{F}$ and $\mathbf{G}$, the set of von Neumann solutions are globally asymptotically stable (Corollary 1). Even in non-strict HCC games, we can add regularization terms to make $L$ strictly convex concave. Small amounts of regularization allows for convergence without significantly perturbing the von Neumann solution (Theorem 7) while increasing regularization enables exponentially faster convergence rates (Theorem 8). Similar to the aforementioned theoretical work, our model of HCC games provides a formal and theoretical tractable testbed for evaluating the performance of different training methods in GAN inspired architectures. As a concrete example, [36] recently proved the success of WGAN training for learning the parameters of non-linearly transformed Gaussian distributions, where for simplicity they replaced the typical Lipschitz constraint of the discriminator function with a quadratic regularizer. Interestingly, we can elucidate on why regularized learning is actually necessary by establishing a formal connection to HCC games. On top of other such ML applications, our game theoretic framework can furthermore capture and generalize evolutionary game theoretic models. [57] analyze a model of evolutionary competition between two species (host-parasite). The outcome of this competition depends on their respective phenotypes (informally their properties, e.g., agility, camouflage, etc.). These phenotypes are encoded via functions that map input vectors (here genotype/DNA sequences) to phenotypes. While [57] proved that learning in these games does not converge to equilibria and typically cycles for almost all initial conditions, we can explicitly construct initial conditions that do not satisfy our definition of safety and end up converging to artificial fixed points. Safety conditions aside, we show that a slight variation of the evolutionary/learning algorithm suffices to resolve the cycling issues and for the dynamics to equilibrate to the von Neumann solution. Hence, we provide the first instance of team zero-sum games [62], a notoriously hard generalization of zero-sum games with a large duality gap, that is solvable by decentralized dynamics.

**Organization.** In Section 2 we provide some preliminary notation, the definition of our model and some useful technical lemmas. Section 3 is devoted to the presentation of our the main results. Section 4 discusses applications of our framework to specific GAN formulations. Section 5 concludes our work with a discussion of future directions and challenges. We defer the full proofs of our results as well as further discussion on applications to the Appendix.

# 2 Preliminaries

## 2.1 Notation

Vectors are denoted in boldface $\mathbf{x}, \mathbf{y}$ unless otherwise indicated are considered as column vectors. We use $\|\cdot\|$ to denote the $\ell_2-$norm. For a function $f : \mathbb{R}^d \to \mathbb{R}$ we use $\nabla f$ to denote its gradient. For functions of two vector arguments, $f(\mathbf{x}, \mathbf{y}) : \mathbb{R}^{d_1} \times \mathbb{R}^{d_2} \to \mathbb{R}$, we use $\nabla_{\mathbf{x}} f, \nabla_{\mathbf{y}} f$ to denote its partial gradient. For the time derivative we will use the dot accent abbreviation, i.e., $\dot{\mathbf{x}} = \frac{d}{dt}[\mathbf{x}(t)]$. A function $f$ will belong to $C^r$ if it is $r$ times continuously differentiable. Additionally, $f \circ g = f(g(\cdot))$ denotes the composition of $f, g$. Finally, the term "sigmoid" function refers to $\sigma : \mathbb{R} \to \mathbb{R}$ such that $\sigma(x) = (1 + e^{-x})^{-1}$.

## 2.2 Hidden Convex Concave Games

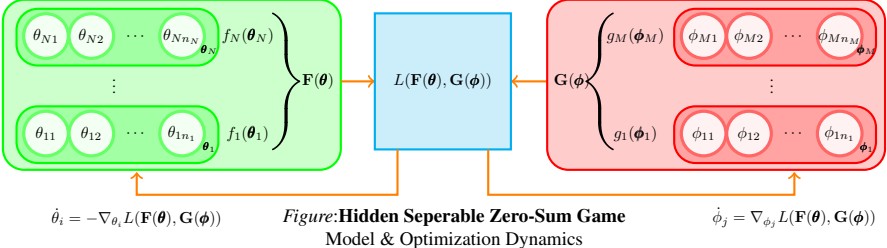

$$\dot{\theta}_i = -\nabla_{\theta_i} L(\mathbf{F}(\boldsymbol{\theta}), \mathbf{G}(\boldsymbol{\phi})) \qquad \qquad \dot{\phi}_j = \nabla_{\phi_j} L(\mathbf{F}(\boldsymbol{\theta}), \mathbf{G}(\boldsymbol{\phi}))$$

*Figure*:**Hidden Seperable Zero-Sum Game**
Model & Optimization Dynamics

We will begin our discussion by defining the notion of convex concave functions as well as strictly convex concave functions. Note that our definition of strictly convex concave functions is a superset of strictly convex strictly concave functions that are usually studied in the literature.

**Definition 1.** $L : \mathbb{R}^n \times \mathbb{R}^m \to \mathbb{R}$ *is convex concave if for every* $\mathbf{y} \in \mathbb{R}^n$ $L(\cdot, \mathbf{y})$ *is convex and for every* $\mathbf{x} \in \mathbb{R}^m$ $L(\mathbf{x}, \cdot)$ *is concave. Function* $L$ *will be called strictly convex concave if it is convex concave and for every* $\mathbf{x} \times \mathbf{y} \in \mathbb{R}^n \times \mathbb{R}^m$ *either* $L(\cdot, \mathbf{y})$ *is strictly convex or* $L(\mathbf{x}, \cdot)$ *is strictly concave.*

At the center of our definition of HCC games is a convex concave utility function $L$. Additionally, each player of the game is equipped with a set of operator functions. The minimization player is equipped with $n$ functions $f_i : \mathbb{R}^{n_i} \to \mathbb{R}$ while the maximization player is equipped with $m$ functions $g_j : \mathbb{R}^{m_j} \to \mathbb{R}$. We will assume in the rest of our discussion that $f_i, g_j, L$ are all $C^2$ functions. The inputs $\boldsymbol{\theta}_i \in \mathbb{R}^{n_i}$ and $\boldsymbol{\phi}_j \in \mathbb{R}^{m_j}$ are grouped in two vectors

$$\boldsymbol{\theta} = \begin{bmatrix} \boldsymbol{\theta}_1 & \cdots & \boldsymbol{\theta}_n \end{bmatrix}^\top \qquad \mathbf{F}(\boldsymbol{\theta}) = \begin{bmatrix} f_1(\boldsymbol{\theta}_1) & \cdots & f_n(\boldsymbol{\theta}_n) \end{bmatrix}^\top$$
$$\boldsymbol{\phi} = \begin{bmatrix} \boldsymbol{\phi}_1 & \cdots & \boldsymbol{\phi}_m \end{bmatrix}^\top \qquad \mathbf{G}(\boldsymbol{\phi}) = \begin{bmatrix} g_1(\boldsymbol{\phi}_1) & \cdots & g_m(\boldsymbol{\phi}_m) \end{bmatrix}^\top$$

We are ready to define the hidden convex concave game

$$(\boldsymbol{\theta}^*, \boldsymbol{\phi}^*) = \arg\min_{\boldsymbol{\theta} \in \mathbb{R}^N} \arg\max_{\boldsymbol{\phi} \in \mathbb{R}^M} L(\mathbf{F}(\boldsymbol{\theta}), \mathbf{G}(\boldsymbol{\phi})).$$

where $N = \sum_{i=1}^n n_i$ and $M = \sum_{j=1}^m m_j$. Given a convex concave function $L$, all stationary points of $L$ are (global) Nash equilibria of the min-max game. We will call the set of all equilibria of $L$, von Neumann solutions of $L$ and denote them by Solution($L$). Unfortunately, Solution($L$) can be empty for games defined over the entire $\mathbb{R}^n \times \mathbb{R}^m$. For games defined over convex compact sets, the existence of at least one solution is guaranteed by von Neumann's minimax theorem. Our definition of HCC games can capture games on restricted domains by choosing appropriately bounded functions $f_i$ and $g_j$. In the following sections, we will just assume that Solution($L$) is not empty. We note that our results hold for both bounded and unbounded $f_i$ and $g_j$. We are now ready to write down the equations of the GDA dynamics for a HCC game:

$$\begin{aligned}
\dot{\boldsymbol{\theta}}_i &= -\nabla_{\boldsymbol{\theta}_i} L(\mathbf{F}(\boldsymbol{\theta}), \mathbf{G}(\boldsymbol{\phi})) &= -\nabla_{\boldsymbol{\theta}_i} f_i(\boldsymbol{\theta}_i) \frac{\partial L}{\partial f_i}(\mathbf{F}(\boldsymbol{\theta}), \mathbf{G}(\boldsymbol{\phi})) \\
\dot{\boldsymbol{\phi}}_j &= \nabla_{\boldsymbol{\phi}_j} L(\mathbf{F}(\boldsymbol{\theta}), \mathbf{G}(\boldsymbol{\phi})) &= \nabla_{\boldsymbol{\phi}_j} g_j(\boldsymbol{\phi}_j) \frac{\partial L}{\partial g_j}(\mathbf{F}(\boldsymbol{\theta}), \mathbf{G}(\boldsymbol{\phi}))
\end{aligned} \tag{1}$$

## 2.3 Reparametrization

The following lemma is useful in studying the dynamics of hidden games.

**Lemma 1.** *Let* $k : \mathbb{R}^d \to \mathbb{R}$ *be a* $C^2$ *function. Let* $h : \mathbb{R} \to \mathbb{R}$ *be a* $C^1$ *function and* $\mathbf{x}(t)$ *denote the unique solution of the dynamical system* $\Sigma_1$. *Then the unique solution for dynamical system* $\Sigma_2$ *is* $\mathbf{z}(t) = \mathbf{x}(\int_0^t h(s) \mathrm{d}s)$

$$\left. \begin{cases} \dot{\mathbf{x}} &= \nabla k(\mathbf{x}) \\ \mathbf{x}(0) &= \mathbf{x}_{init} \end{cases} \right\} : \Sigma_1 \qquad \left. \begin{cases} \dot{\mathbf{z}} &= h(t) \nabla k(\mathbf{z}) \\ \mathbf{z}(0) &= \mathbf{x}_{init} \end{cases} \right\} : \Sigma_2 \tag{2}$$

By choosing $h(t) = -\partial L(\mathbf{F}(t), \mathbf{G}(t)) / \partial f_i$ and $h(t) = \partial L(\mathbf{F}(t), \mathbf{G}(t)) / \partial g_j$ respectively, we can connect the dynamics of each $\boldsymbol{\theta}_i$ and $\boldsymbol{\phi}_j$ under Equation (1) to gradient ascent on $f_i$ and $g_j$. Applying Lemma 1, we get that trajectories of $\boldsymbol{\theta}_i$ and $\boldsymbol{\phi}_j$ under Equation (1) are restricted to be subsets of the corresponding gradient ascent trajectories with the same initializations. For example, in Figure 1 $\theta_i(t)$ can not escape the purple section if it is initialized at (a) neither the orange section if it is initialiazed at (f). This limits the attainable values that $f_i(t)$ and $g_j(t)$ can take for a specific initialization. Let us thus define the following:

**Definition 2.** *For each initialization* $\mathbf{x}(0)$ *of* $\Sigma_1$, $\mathrm{Im}_k(\mathbf{x}(0))$ *is the image of* $k \circ \mathbf{x} : \mathbb{R} \to \mathbb{R}$.

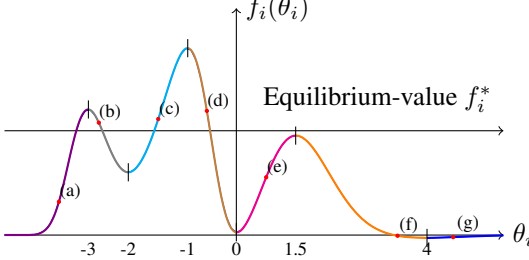

Figure 1: Neither Gradient Descent nor Ascent can traverse stationary points. An immediate consequence of Lemma 1 is that if we initialize in the above example $\theta_i(0)$ at (a), $f_i(\theta_i(t))$ can not escape the purple section. This extends to cases where $\boldsymbol{\theta}_i$ is vector of variables.

Applying Definition 2 in the above example, $\mathrm{Im}_{f_i}(\theta_i(0)) = (f_i(-2), f_i(-1))$ if $\theta_i$ is initialized at (c). Additionally, observe that in each colored section $f_i(\theta_i(t))$ uniquely identifies $\theta_i(t)$. Generally, even in the case that $\boldsymbol{\theta}_i$ are vectors, Lemma 1 implies that for a given $\boldsymbol{\theta}_i(0)$, $f_i(\boldsymbol{\theta}_i(t))$ uniquely identifies $\boldsymbol{\theta}_i(t)$. As a result we get that a new dynamical system involving only $f_i$ and $g_j$

**Theorem 1.** *For each initialization $(\boldsymbol{\theta}(0), \boldsymbol{\phi}(0))$ of Equation* (1)*, there are $C^1$ functions $X_{\boldsymbol{\theta}_i(0)}$, $X_{\boldsymbol{\phi}_j(0)}$ such that $\boldsymbol{\theta}_i(t) = X_{\boldsymbol{\theta}_i(0)}(f_i(t))$ and $\boldsymbol{\phi}_j(t) = X_{\boldsymbol{\phi}_j(0)}(g_j(t))$. If $(\boldsymbol{\theta}(t), \boldsymbol{\phi}(t))$ satisfy Equation* (1) *then $f_i(t) = f_i(\boldsymbol{\theta}_i(t))$ and $g_j(t) = g_j(\boldsymbol{\phi}_j(t))$ satisfy*

$$\dot{f}_i = -\|\nabla_{\boldsymbol{\theta}_i} f_i(X_{\boldsymbol{\theta}_i(0)}(f_i))\|^2 \frac{\partial L}{\partial f_i}(\mathbf{F}, \mathbf{G})$$

$$\dot{g}_j = \|\nabla_{\boldsymbol{\phi}_j} g_j(X_{\boldsymbol{\phi}_j(0)}(g_j))\|^2 \frac{\partial L}{\partial g_j}(\mathbf{F}, \mathbf{G})$$

(3)

By determining the ranges of $f_i$ and $g_j$, an initialization clearly dictates if a von Neumann solution is attainable. In Figure 1 for example, any point of the pink, orange or blue colored section like (e), (f) or (g) can not converge to a von Neumann solution with $f_i(\theta_i) = f_i^*$. The notion of *safety* captures which initializations can converge to a given element of Solution($L$).

**Definition 3.** *. We will call the initialization $(\boldsymbol{\theta}(0), \boldsymbol{\phi}(0))$ safe for a $(\mathbf{p}, \mathbf{q}) \in$ Solution($L$) if $\boldsymbol{\phi}_i(0)$ and $\boldsymbol{\theta}_j(0)$ are not stationary points of $f_i$ and $g_j$ respectively and $p_i \in \mathrm{Im}_{f_i}(\boldsymbol{\theta}_i(0))$ and $q_j \in \mathrm{Im}_{g_j}(\boldsymbol{\phi}_j(0))$.*

Leveraging the Center-Stable Manifold Theorem [55], the following observation shows that under mild assumptions almost all initializations are safe:

**Theorem 2.** *If $f_i$ and $g_j$ have isolated stationary points, only strict saddle points, compact sublevel-sets, both equilibria $p_i \in (\max \mathrm{LocalMin}(f_i), \min \mathrm{LocalMax}(f_i))$ and $q_j \in (\max \mathrm{LocalMin}(g_j), \min \mathrm{LocalMax}(g_j))$, then almost all initializations are safe for a $(\mathbf{p}, \mathbf{q}) \in$ Solution($L$).*

Finally, in the following sections we use some fundamental notions of stability. We call an equilibrium $\mathbf{x}^*$ of an autonomous dynamical system $\dot{\mathbf{x}} = \mathcal{D}(\mathbf{x}(t))$ *stable* if for every neighborhood $U$ of $\mathbf{x}^*$ there is a neighborhood $V$ of $\mathbf{x}^*$ such that if $\mathbf{x}(0) \in V$ then $\mathbf{x}(t) \in U$ for all $t \geq 0$. We call a set $S$ *asymptotically stable* if there exists a neighborhood $\mathcal{R}$ such that for any initialization $\mathbf{x}(0) \in \mathcal{R}$, $\mathbf{x}(t)$ approaches $S$ as $t \to +\infty$. If $\mathcal{R}$ is the whole space the set *globally asymptotically stable*.

## 3 Learning in Hidden Convex Concave Games

### 3.1 General Case

Our main results are based on designing a Lyapunov function for the dynamics of Equation (3):

**Lemma 2.** *If $L$ is convex concave and $(\boldsymbol{\phi}(0), \boldsymbol{\theta}(0))$ is a safe for $(\mathbf{p}, \mathbf{q}) \in$ Solution($L$), then the following quantity is non-increasing under the dynamics of Equation* (3)*:*

$$H(\mathbf{F}, \mathbf{G}) = \sum_{i=1}^{N} \int_{p_i}^{f_i} \frac{z - p_i}{\|\nabla f_i(X_{\boldsymbol{\theta}_i(0)}(z))\|^2} \mathrm{d}z + \sum_{j=1}^{M} \int_{q_j}^{g_j} \frac{z - q_j}{\|\nabla g_j(X_{\boldsymbol{\phi}_j(0)}(z))\|^2} \mathrm{d}z \qquad (4)$$

Observe that our Lyapunov function here is not the distance to $(\mathbf{p}, \mathbf{q})$ as in a classical convex concave game. The gradient terms account for the non constant multiplicative terms in Equation (3). Indeed if the game was not hidden and $f_i$ and $g_j$ were the identity functions then $H$ would coincide with the Euclidean distance to $(\mathbf{p}, \mathbf{q})$. Our first theorem employs the above Lyapunov function to show that $(\mathbf{p}, \mathbf{q})$ is stable for Equation (3).

**Theorem 3.** *If $L$ is convex concave and $(\boldsymbol{\phi}(0), \boldsymbol{\theta}(0))$ is a safe for $(\mathbf{p}, \mathbf{q}) \in$ Solution($L$), then $(\mathbf{p}, \mathbf{q})$ is stable for Equation* (3).

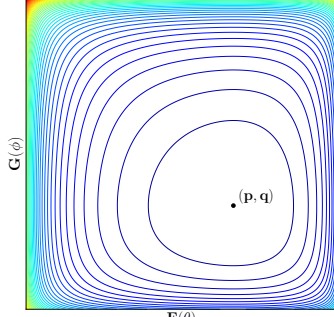

Figure 2: Level sets of Lyapunov function of Equation (4) for both $\mathbf{F}$ and $\mathbf{G}$ being one dimensional sigmoid functions.

Clearly, for the special case of globally invertible
functions $\mathbf{F}, \mathbf{G}$ we could come up with an equivalent Lyapunov function in the $\boldsymbol{\theta}, \boldsymbol{\phi}$-space. In this case it is straightforward to transfer the stability results from the induced dynamical system of $\mathbf{F}, \mathbf{G}$ (Equation (3)) to the initial dynamical system of $\boldsymbol{\theta}, \boldsymbol{\phi}$ (Equation (1)). For example we can prove the following result:

**Theorem 4.** *If $f_i$ and $g_j$ are sigmoid functions and $L$ is convex concave and there is a $(\boldsymbol{\phi}(0), \boldsymbol{\theta}(0))$ that is safe for $(\mathbf{p}, \mathbf{q}) \in Solution(L)$, then $(\mathbf{F}^{-1}(\mathbf{p}), \mathbf{G}^{-1}(\mathbf{q}))$ is stable for Equation (1).*

In the general case though, stability may not be guaranteed in the parameter space of Equation (1). We will instead prove a weaker notion of stability, which we call hidden stability. Hidden stability captures that if $(\mathbf{F}(\boldsymbol{\theta}(0)), \mathbf{G}(\boldsymbol{\phi}(0)))$ is close to a von Neumann solution, then $(\mathbf{F}(\boldsymbol{\theta}(t)), \mathbf{G}(\boldsymbol{\phi}(t)))$ will remain close to that solution. Even though hidden stability is weaker, it is essentially what we are interested in, as the output space determines the utility that each player gets. Here we provide sufficient conditions for hidden stability.

**Theorem 5** (Hidden Stability). *Let $(\mathbf{p}, \mathbf{q}) \in Solution(L)$. Let $R_{f_i}$ and $R_{g_j}$ be the set of regular values[1] of $f_i$ and $g_j$ respectively. Assume that there is a $\xi > 0$ such that $[p_i - \xi, p_i + \xi] \subseteq R_{f_i}$ and $[q_j - \xi, q_j + \xi] \subseteq R_{g_j}$. Define*

$$r(t) = \|\mathbf{F}(\boldsymbol{\theta}(t)) - \mathbf{p}\|^2 + \|\mathbf{G}(\boldsymbol{\phi}(t)) - \mathbf{q}\|^2.$$

*If $f_i$ and $g_j$ are proper functions[2], then for every $\epsilon > 0$, there is an $\delta > 0$ such that*

$$r(0) < \delta \implies \forall t \geq 0 : r(t) < \epsilon.$$

Unfortunately hidden stability still does not imply convergence to von Neumann solutions. [65] studied hidden bilinear games and proved that $\dot{H} = 0$ for this special class of HCC games. Hence, a trajectory is restricted to be a subset of a level set of $H$ which is bounded away from the equilibrium as shown in Figure 2. To sidestep this, we will require in the next subsection the hidden game to be strictly convex concave.

## 3.2 Hidden strictly convex concave games

In this subsection we focus on the case where $L$ is a strictly convex concave function. Based on Definition 1, a strictly convex concave game is not necessarily strictly convex strictly concave and thus it may have a continuum of von Neumann solutions. Despite this, LaSalle's invariance principle, combined with the strict convexity concavity, allows us to prove that if $(\boldsymbol{\theta}(0), \boldsymbol{\phi}(0))$ is safe for $Z \subseteq Solution(L)$ then $Z$ is locally asymptotically stable for Equation (3).

**Lemma 3.** *Let $L$ be strictly convex concave and $Z \subset Solution(L)$ is the non empty set of equilbria of $L$ for which $(\boldsymbol{\theta}(0), \boldsymbol{\phi}(0))$ is safe. Then $Z$ is locally asymptotically stable for Equation (3).*

The above lemma however does not suffice to prove that for an arbitrary initialization $(\boldsymbol{\theta}(0), \boldsymbol{\phi}(0))$, $(\mathbf{F}(t), \mathbf{G}(t))$ approaches $Z$ as $t \to +\infty$. In other words, *a-priori* it is unclear if $(\mathbf{F}(\boldsymbol{\theta}(0)), \mathbf{G}(\boldsymbol{\phi}(0)))$ is necessarily inside the region of attraction (ROA) of $Z$. To get a refined estimate of the ROA of $Z$, we analyze the behavior of $H$ as $f_i$ and $g_j$ approach the boundaries of $Im_{f_i}(\boldsymbol{\theta}_i(0))$ and $Im_{g_j}(\boldsymbol{\phi}_j(0))$ and more precisely we show that the level sets of $H$ are bounded. Once again the corresponding analysis is trivial for convex concave games, since the level sets are spheres around the equilibria.

**Theorem 6.** *Let $L$ be strictly convex concave and $Z \subset Solution(L)$ is the non empty set of equilbria of $L$ for which $(\boldsymbol{\theta}(0), \boldsymbol{\phi}(0))$ is safe. Under the dynamics of Equation (1) $(\mathbf{F}(\boldsymbol{\theta}(t)), \mathbf{G}(\boldsymbol{\theta}(t)))$ converges to a point in $Z$ as $t \to \infty$.*

The theorem above guarantees convergence to a von Neumann solution for all initializations that are safe for at least one element of $Solution(L)$. However, this is not the same as global asymptotic stability. To get even stronger guarantees, we can assume that all initializations are safe. In this case it is straightforward to get a global asymptotic stability result:

**Corollary 1.** *Let $L$ be strictly convex concave and assume that all intitializations are safe for at least one element of Solution(L). The following set is globally asymptotically stable for continuous GDA dynamics.*

$$\{(\boldsymbol{\theta}^*, \boldsymbol{\phi}^*) \in \mathbb{R}^n \times \mathbb{R}^m : (F(\boldsymbol{\theta}^*), G(\boldsymbol{\phi}^*)) \in Solution(L)\}$$

---

[1]A value $a \in Im\, f$ is called a regular value of $f$ if $\forall q \in dom\, f : f(q) = a$, it holds $\nabla f(q) \neq \mathbf{0}$.

[2]A function is proper if inverse images of compact subsets are compact.

Notice that the above approach on global asymptotic convergence using Lyapunov arguments can be extended to other popular alternative gradient-based heuristics like variations of Hamiltonian Gradient Descent. For concision, we defer the exact statements, proofs in the supplement.

### 3.3 Convergence via regularization

Regularization is a key technique that works both in the practice of GANs [47, 33] and in the theory of convex concave games [56, 59, 60]. Our settings of hidden convex concave games allows for provable guarantees for regularization in a wide class of settings, bringing closer practical and theoretical guarantees. Let us have a utility $L(\mathbf{x}, \mathbf{y})$ that is convex concave but not strictly. Here we will propose a modified utility $L'$ that is strictly convex strictly concave. Specifically we will choose

$$L'(\mathbf{x}, \mathbf{y}) = L(\mathbf{x}, \mathbf{y}) + \frac{\lambda}{2}\|\mathbf{x}\|^2 - \frac{\lambda}{2}\|\mathbf{y}\|^2$$

The choice of the parameter $\lambda$ captures the trade-off between convergence to the original equilibrium of $L$ and convergence speed. On the one hand, invoking the implicit function theorem, we get that for small $\lambda$ the equilibria of $L$ are not significantly perturbed.

**Theorem 7.** *If $L$ is a convex concave function with invertible Hessians at all its equilibria, then for each $\epsilon > 0$ there is a $\lambda > 0$ such that $L'$ has equilibria that are $\epsilon$-close to the ones of $L$.*

Note that invertibility of the Hessian means that $L$ must have a unique equilibrium. On the other hand increasing $\lambda$ increases the rate of convergence of safe initializations to the perturbed equilibrium.

**Theorem 8.** *Let $(\boldsymbol{\theta}(0), \boldsymbol{\phi}(0))$ be a safe initialization for the unique equilibrium of $L'$ $(\mathbf{p}, \mathbf{q})$. If*

$$r(t) = \|\mathbf{F}(\boldsymbol{\theta}(t)) - \mathbf{p}\|^2 + \|\mathbf{G}(\boldsymbol{\phi}(t)) - \mathbf{q}\|^2$$

*then there are initialization dependent constants $c_0, c_1 > 0$ such that $r(t) \leq c_0 \exp(-\lambda c_1 t)$.*

## 4 Applications

In this section, we discuss how HCC framework can be used to give new insights in a variety of application areas including min-max training for GANs and Evolutionary Game Theory. We also describe applications of regularization to normal form zero sum games in Appendix D.3.

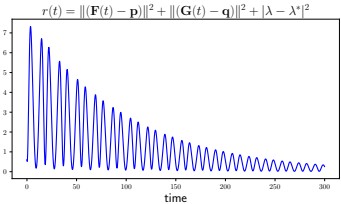
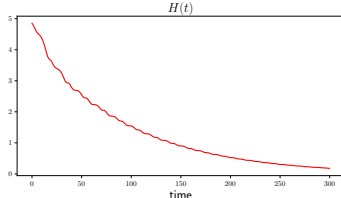

Figure 3: Both the $\ell_2$ distance from the equilibrium and $H(t)$ converge to zero but only the latter does so monotonically For $p_{\text{data}}$ we choose a fully mixed distribution of dimension $d = 4$. Given the sigmoid activations all the initializations are safe. We defer the detailed proof of convergence in Appendix D.2.

**Hidden strictly convex-concave games**. We will start our discussion with the fundamental generative architecture of [26]'s GAN. In the *vanilla GAN* architecture, as it is commonly referred, our goal is to find a generator distribution $p_G$ that is close to an input data distribution $p_{\text{data}}$. To find such a generator function, we can use a discriminator $D$ that "criticizes" the deviations of the generator from the input data distribution. For the case of a discrete $p_{\text{data}}$ over a set $\mathcal{N}$, the minimax problem of [26] is the following:

$$\min_{\substack{p_G(x) \geq 0, \\ \sum_{x \in \mathcal{N}} p_G(x) = 1}} \max_{D \in (0,1)^{|\mathcal{N}|}} V(G, D)$$

where $V(G, D) = \sum_{x \in \mathcal{N}} p_{\text{data}}(x) \log(D(x)) + \sum_{x \in \mathcal{N}} p_G(x) \log(1 - D(x))$. The problem above can be formulated as a constrained strictly convex-concave hidden game. On the one hand, for a fixed discriminator $D^*$, the $V(G, D^*)$ is linear over the $p_G(x)$. On the other hand, for a fixed generator $G^*$,

$V(G^*, D)$ is strongly-concave. We can implement the inequality constraints on both the generator probabilities and discriminator using sigmoid activations. For the equality constraint $\sum_{x \in \mathcal{N}} p_{\mathrm{G}}(x) = 1$ we can introduce a Langrange multiplier. Having effectively removed the constraints, we can see in Figure 3, the dynamics of Equation (1) converge to the unique equilibrium of the game, an outcome consistent with our results in Corollary 1. While the Euclidean distance to the equilibrium is not monotonically decreasing, $H(t)$ is.

**Hidden convex-concave games & Regularizaiton**. An even more interesting case is Wassertein GANs–WGANs [4]. One of the contributions of [36] is to show that WGANs trained with Stochastic GDA can learn the parameters of Gaussian distributions whose samples are transformed by non-linear activation functions. It is worth mentioning that the original WGAN formulation has a Lipschitz constraint in the discriminator function. For simplicity, [36] replaced this constraint with a quadratic regularizer. The min-max problem for the case of one-dimensional Gaussian $\mathcal{N}(0, \alpha_*^2)$ and linear discriminator $D_v(x) = v^\top x$ with $x^2$ activation is:

$$\min_{\alpha \in \mathbb{R}} \max_{v \in \mathbb{R}} V_{\mathrm{WGAN}}(G_\alpha, D_v) \qquad = \mathbb{E}_{\mathbf{X} \sim p_{\mathrm{data}}}[D(\mathbf{X})] - \mathbb{E}_{\mathbf{X} \sim p_{\mathrm{G}}}[D(\mathbf{X})] - v^2/2$$

$$= \mathbb{E}_{x \sim \mathcal{N}(0, \alpha_*^2)^2}[vx] - \mathbb{E}_{x \sim \mathcal{N}(0, \alpha^2)^2}[vx] - v^2/2$$

$$= (\alpha_*^2 - \alpha^2)v - v^2/2$$

Observe that $V_{\mathrm{WGAN}}$ is not convex-concave but it can posed as a hidden strictly convex-concave game with $\mathbf{G}(\alpha) = (\alpha_*^2 - \alpha^2)$ and $\mathbf{F}(v) = v$. When computing expectations analytically without sampling, Theorem 6 guarantees convergence. In contrast, without the regularizer $V_{\mathrm{WGAN}}$ can be modeled as a hidden bilinear game and thus GDA dynamics cycle. Empirically, these results are robust to discrete and stochastic updates using sampling as shown in Figure 4. Therefore regularization in the work of [36] was a vital ingredient in their proof strategy and not just an implementation detail.

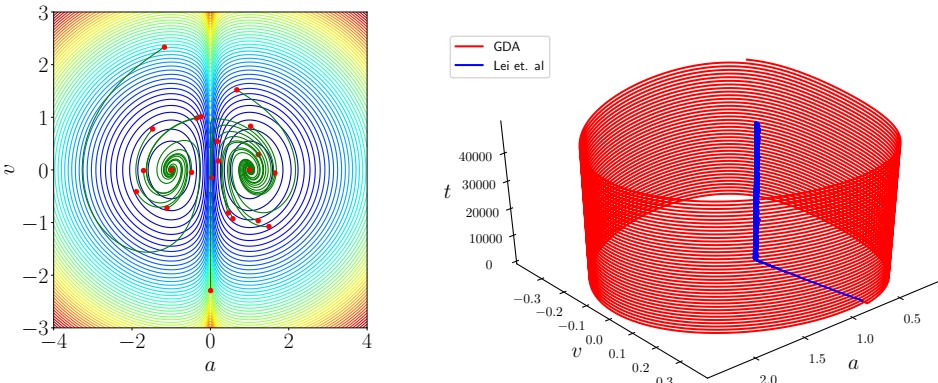

Figure 4: On the left, we show the trajectories of regularized GDA for $\alpha_*^2 = 1$ as well as the level sets of Equation (4). All trajectories (green curves-initialized at the red points) converge to one of the two equilibria $(0, 1)$ and $(0, -1)$ whereas without regularization, GDA would cycle on the level sets. In the right figure, we replace the exact expectations in $V_{\mathrm{WGAN}}$ with approximations via sampling and continuous time updates on $\alpha$ and $v$ with discrete ones. For small learning rates and large sample sizes, unregularized GDA continues to cycle. In contrast, the regularization approach of [36] converges to the $(0, 1)$ equilibrium.

The two applications of HCC games in GANs are not isolated findings but instances of a broader pattern that connects HCC games and standard GAN formulations. As noted by [27], if updates in GAN applications were directly performed in the "functional space", i.e. the generator and discriminator outputs, then standard arguments from convex concave optimization would imply convergence to global Nash equilibria. Indeed, standard GAN formulations like the vanilla GAN [26], f-GAN [50] and WGAN [4] can all be thought of as convex concave games in the space of generator and discriminator outputs. Given that the connections between convex concave games and standard GAN objectives in the output space is missing from recent literature, in Appendix D.1 we show how one can apply Von Neumann's minimax theorem to derive the optimal generators and discriminators even in the non-realizable case. In practice, the updates happen in the parameter space and thus convexity arguments no longer apply. Our study of HCC games is a stepping stone towards bridging the gap in convergence guarantees between the case of direct updates in the output space and the parameter space.

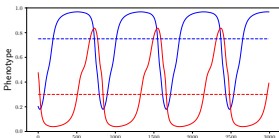 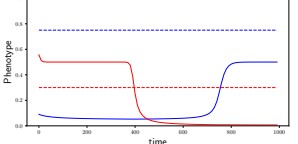 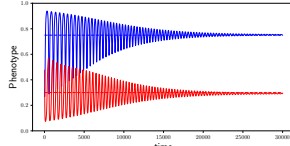

Figure 5: The above figures describes the evolution of the expected phenotype for two species A,B. The left one corresponds to a safe initialization leading to periodic trajectories. The middle one corresponds to an unsafe initialization where $\theta_1(0) = \theta_2(0)$. The dynamics converge albeit to a spurious equilibrium, that is different from the hidden game equilibrium (dash lines). Finally, the right one corresponds to a safe initialization of the regularized game, which converges to a slightly perturbed equilibrium (Theorem 7).

**Evolutionary Game Theory & Biology**. The study of learning dynamics in games has always been strongly and inherently connected with mathematical models of biology and evolution. Typically, this line of research is studied under the name of *Evolutionary Game Theory* [28, 67]. Zero-sum games and variants thereof are of particular interest for this line of work as they encode settings of direct competition between species (e.g., prey-predator or host-parasite/virus). Even in the simplest such setting of matrix zero-sum games, used to capture competition between asexually reproducing species, it is well known that the emerging dynamics can be non-equilibrating and even chaotic [61, 58].

Studying the effects of evolutionary competition between sexually evolving species results in significantly more intricate models, as it does not suffice to merely keep track of the fractions of the different types of individuals that self-replicate. Instead it is necessary to keep a much more detailed accounting of the evolution of the frequencies of different genes that get reshuffled and recombined to create new individuals, whilst giving evolutionary preference to the most fit individuals given the current environment. Recent work on intersection of learning theory and game theory has provided concrete such game theoretic models [37, 14, 44, 42]. Due to the intricate nature of their dynamics, deciding even the simplest questions in regards to them (e.g. does genetic diversity survive or not?) is typically computationally hard [43].

A notable exception, where the dynamics of sexual evolution and, in fact, sexual competition have been relatively thoroughly understood, is the work of [57], on two species (host-parasite) antagonism. The outcome of this competition depends on their respective phenotypes (informally their properties, e.g., large wings versus small wings.) of the two species. The crucial assumption that makes this model theoretically tractable is that the phenotype for each species is a Boolean attribute (this assumption is also used [38]). Despite these simplifications, the dynamics are still *not equilibrating* and are, in fact, cyclic for almost all initial conditions. Two natural questions emerge: 1) *Is the almost everywhere condition necessary? I.e. Do there exist initial conditions which are not cyclic?* 2) *More importantly, can a slightly perturbed dynamic stabilize these systems and converge to a meaningful equilibrium?* Next, we will see how our framework addresses both of these questions.

To understand the connection these we will examine the model of [57] in more detail. Concretely, the phenotype of species $A, B$ can be described as a Boolean function over the species genome which is encoded by a binary string (this acts as a simplified version of a DNA string). While the *phenotype* plays the dominant role for the survival of the species, sexual reproduction modifies only the *genotype* of an organism. As a result the species are actually involved in a hidden zero-sum game. More formally, each species is game-theoritically represented as a team of agents where each agent controls one bit of the genotype:

$$\mathsf{G}_A = (\mathsf{g}_1^A, \cdots, \mathsf{g}_n^A), \mathsf{G}_B = (\mathsf{g}_1^B, \cdots, \mathsf{g}_m^B)$$
$$\mathsf{u}_A = L[\mathsf{Phenotype}_A(\mathsf{G}_A), \mathsf{Phenotype}_B(\mathsf{G}_B)]$$
$$\mathsf{u}_B = -\mathsf{u}_A$$

Where $\mathsf{g}_i^A, \mathsf{g}_j^B \in \{0, 1\}$, $\mathsf{Phenotype}_A, \mathsf{Phenotype}_B$ is a Boolean function (e.g., AND, XOR) and $L$ is a $2 \times 2$ matrix encoding a zero-sum game (e.g., Matching Pennies). Naturally, one can allow agents to use randomized/mixed strategies in which case the expected utilities of all agents/genes are defined using the standard multi-linear extension of utilities. Thus, these models of evolutionary sexual competition share the same basic structure as hidden linear-linear games, which explains their recurrent, non-equilibrating nature [65].

In Figure 5, each gene/agent $g_i^A$ tunes one real variable $\theta_i$ such that $\Pr[g_i^A = 1] = \sigma(\theta_i)$ and gene/agent $g_j^B$ tunes one real variable $\phi_j$ correspondingly. Choosing as Boolean phenotype to be the XOR of two genes, almost all initializations are safe for any bilinear game with a mixed equilibrium. Actually, only the case $\theta_1(0) = \theta_2(0)$ or $\phi_1(0) = \phi_2(0)$ can be problematic, since for XOR the expected phenotype is bounded in $[0, 0.5]$ and a mixed equilibrium out of this range would be infeasible. Finally, leveraging Theorem 7, we can design a regularized version of the game such that the dynamics converge arbitrarily close to the true von Neumann solution of these games, which is encoded by the min-max strategies of the hidden bi-linear zero-sum game.

## 5 Discussion & Future Work

While this work is a promising first step towards understanding GAN training, significant challenges remain. Neural network architectures do not use disjoint set of parameters for each of the outputs. Additionally, the hidden competition of GANs can take place in an output space of probability distributions and classifiers whose vector space dimension is typically infinite. On the bright side, we establish point-wise (day to day) convergence results which are, to the best of our knowledge, the first result of their kind for a wide class of non-convex non-concave games that do not necessarily satisfy the Polyak-Łojasiewicz conditions studied in [68]. Such conditions imply that the notions of saddle points, global min-max and stationary points coincide. Instead our work showcases how to make progress without leveraging such strong assumptions in zero-sum games. Beyond ML applications, we believe that our framework could provide even further insights for evolutionary game theory, mathematical biology as well as team-zero-sum games. For example an interesting hybrid class of games could be network generalizations of team-zero-sums games, e.g. by combining [12] and [57].

### Acknowledgments and Disclosure of Funding

This research/project is supported in part by the National Research Foundation, Singapore under its AI Singapore Program (AISG Award No: AISG2-RP-2020-016), NRF 2018 Fellowship NRF-NRFF2018-07, NRF2019-NRF-ANR095 ALIAS grant, grant PIE-SGP-AI-2018-01, AME Programmatic Fund (Grant No. A20H6b0151) from the Agency for Science, Technology and Research (A*STAR). Additionally, E.V. Vlatakis-Gkaragkounis is grateful to be supported by NSF grants CCF-1703925, CCF1763970, CCF-1814873, CCF-1563155, and by the Simons Collaboration on Algorithms and Geometry. He would like to acknowledge the support of Onassis Foundation under the Scholarship(ID: FZN 010-1/2017-2018.)

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
