# A   Background

## A.1   Background in dynamical systems

Our analysis combines tools from dynamical systems, stability analysis and invariance principles theory. We start with the definitions of the different stability notions. We remind the well known Lyapunov's Lyapunov stability criterion (**Theorem 9**) Stability analysis in convex concave games is further complicated due to the possibility of non-isolated fixed points. To tackle this issue, we recall Krasovskii-LaSalle's Invariance Principle (**Theorem 10**), a powerful result that has several implications for the asymptotic stability of a set in an autonomous (possibly nonlinear) dynamical system. In the special case where the goal set contains only stable fixed points a pointwise convergence theorem can be derived (**Theorem 11**). The Center-Stable Manifold Theorem (**Theorem 12**) is going to be a key ingredient of the proof of Theorem 2. Finally, we remind the notions of diffeomorphism and topological conjugacy of two dynamical systems, which are useful to transfer behavioral claims between equivalent dynamics.

Let $\mathbf{f} : D \to \mathbb{R}^n$ be a locally Lipschitz map from a domain $D \subset \mathbb{R}^n$ to $\mathbb{R}^n$. We consider dynamical systems of the form

$$\dot{\mathbf{x}} = \mathbf{f}(\mathbf{x}) \qquad\qquad (\star)$$

A point $\bar{\mathbf{x}}$ for which $\mathbf{f}(\bar{\mathbf{x}}) = \mathbf{0}$ is called a fixed point. We will be interested in the following notions of stability for the fixed point points of Equation $(\star)$.

**Definition 4** (Stability properties, [32, Definition 4.1]). *The fixed point $\mathbf{x} = \mathbf{0}$ of Equation $(\star)$ is*

- *stable if, for each $\epsilon > 0$, there is a $\delta = \delta(\epsilon) > 0$ such that*

$$\|\mathbf{x}(0)\| < \delta \implies \|\mathbf{x}(t)\| < \epsilon \quad \forall t \geq 0$$

- *unstable if it is not stable*

- *asymptotically stable if it is stable and $\delta$ can be chosen such that*

$$\|\mathbf{x}(0)\| < \delta \implies \lim_{t \to \infty} \mathbf{x}(t) = \mathbf{0}$$

The Lyapunov Theorem will be a useful tool to prove (asymptotic) stability of a fixed point.

**Theorem 9** (Lyapunov Theorem, [32, Theorem 4.1]). *Let $\mathbf{x} = \mathbf{0}$ be a fixed point for Equation $(\star)$ and $D \subset \mathbb{R}^n$ be a domain containing $\mathbf{x} = \mathbf{0}$. Let $V : D \to \mathbb{R}$ be a continuously differentiable function such that*

$$V(\mathbf{0}) = 0 \text{ and } V(\mathbf{x}) > 0 \text{ in } D - \{\mathbf{0}\}$$

$$\dot{V}(\mathbf{x}) \leq 0 \text{ in } D$$

*then $\mathbf{x} = \mathbf{0}$ is stable. Moreover if*

$$\dot{V}(\mathbf{x}) < 0 \text{ in } D - \{\mathbf{0}\}$$

*then $\mathbf{x} = \mathbf{0}$ is asymptotically stable.*

Unfortunately, the Lyapunov theorem is not very helpful when it comes to proving convergence in dynamical systems with non isolated fixed points. By definition, non-isolated fixed points cannot be asymptotically stable. Non isolated fixed points may give rise to more complex behavior than point-wise convergence.

**Definition 5.** *We say that a trajectory $\mathbf{x}(t)$ approaches a set $\mathcal{M}$ as $t \to \infty$ if for each $\epsilon > 0$ there is a $T > 0$ such that*

$$dist(\mathbf{x}(t), \mathcal{M}) < \epsilon, \quad \forall t > T$$

*where the operator "dist" is the minimum distance from a point to a set $\mathcal{M}$*

$$dist(\mathbf{p}, \mathcal{M}) = \inf_{\mathbf{x} \in \mathcal{M}} \|\mathbf{p} - \mathbf{x}\|$$

**Definition 6.** *We say that a set $\mathcal{M}$ is invariant for Equation $(\star)$ if*

$$\mathbf{x}(0) \in \mathcal{M} \implies \mathbf{x}(t) \in \mathcal{M}, \quad \forall t \in \mathbb{R}$$

*We will say $\mathcal{M}$ is positively invariant if the above holds for $t \geq 0$.*

We are ready to state LaSalle's Invariance Principle, a general theorem that can help us study the stability of non isolated fixed points.

**Theorem 10** ( LaSalle's Invariance Principle, [32, Theorem 4.4]). *Let $\Omega \subset D$ be a compact set that is positively invariant with respect to Equation $(\star)$. Let $V : D \to \mathbb{R}$ be a continuously differentiable function such that $\dot{V}(\mathbf{x}) \leq 0$ in $\Omega$. Let $E$ be the set of all points where $\dot{V}(\mathbf{x}) = 0$. Let $\mathcal{M}$ be the largest invariant set in $E$. Then every solution starting in $\Omega$ approaches $\mathcal{M}$ as $t \to \infty$.*

LaSalle's theorem does not give us pointwise convergence directly. But in the special case that $\mathcal{M}$ contains only stable fixed points we can apply the following theorem

**Theorem 11** (Pointwise Convergence Theorem, [10, Proposition 5.4]). *Let $\mathbf{x}(t)$ be a trajectory of Equation $(\star)$. If the positive limit sets of $\mathbf{x}(t)$ contain a stable fixed point then $\mathbf{x}(t)$ converges to it as $t \to \infty$.*

**Definition 7** (Differomorphism, [55]). *Let $U, V$ be manifolds. A map $f : U \to V$ is called a diffeomorphism if $f$ carries $U$ onto $V$ and also both $f$ and $f^{-1}$ are smooth.*

**Definition 8** (Topological conjugacy, [55]). *Two flows $\Phi_t : A \to A$ and $\Psi_t : B \to B$ are conjugate if there exists a homeomorphism $g : A \to B$ such that*

$$\forall \boldsymbol{x} \in A, t \in \mathbb{R} : g(\Phi_t(\boldsymbol{x})) = \Psi_t(g(\boldsymbol{x}))$$

*Furthermore, two flows $\Phi_t : A \to A$ and $\Psi_t : B \to B$ are diffeomorphic if there exists a diffeomorphism $g : A \to B$ such that*

$$\forall \boldsymbol{x} \in A, t \in \mathbb{R} : g(\Phi_t(\boldsymbol{x})) = \Psi_t(g(\boldsymbol{x})).$$

*If two flows are diffeomorphic, then their vector fields are related by the derivative of the conjugacy. That is, we get precisely the same result that we would have obtained if we simply transformed the coordinates in their differential equations.*

**Theorem 12** (Stable Manifold Theorem for Continuous Time Dynamical Systems p.120 [55]). *Let $E$ be an open subset of $\mathbb{R}^n$ containing the origin, let $f \in C^1(E)$, and let $\phi_t$ be the flow of the nonlinear system $\dot{\boldsymbol{x}} = f(\boldsymbol{x})$. Suppose that $f(\mathbf{0}) = \mathbf{0}$ and that $Df(\mathbf{0})$ has $k$ eigenvalues with negative real part and $n - k$ eigenvalues with positive real part. Then there exists a $k$-dimensional differentiable manifold $S$ tangent to the stable subspace $E^s$ of the linear system $\dot{\boldsymbol{x}} = Df(\mathbf{0})\boldsymbol{x}$ at $\mathbf{0}$ such that for all $t \geq 0$, $\phi_t(S) \subseteq S$ and for all $\boldsymbol{x}_0 \in S$:*

$$\lim_{t \to \infty} \phi_t(\boldsymbol{x}_0) = \mathbf{0}$$

*and there exists an $n - k$ dimensional differentiable manifold $U$ tangent to the unstable subspace $E^u$ of the linear system $\dot{\boldsymbol{x}} = Df(\mathbf{0})\boldsymbol{x}$ at $\mathbf{0}$ such that for all $t \leq 0$, $\phi_t(U) \subseteq U$ and for all $\boldsymbol{x}_0 \in U$:*

$$\lim_{t \to -\infty} \phi_t(\boldsymbol{x}_0) = \mathbf{0}$$

**Remark 1.** *While the focus of our work relies on results in continuous dynamics, for the interested reader the broader possible implications of our results in discrete time algorithms should be highlighted. Indeed, under some technical conditions described in work of Benaim [9] , whose verification for HCC games lies beyond the scope of the current work, one could argue that discretized dynamics, even the stochastic ones, have the same convergence behaviour as their continuous counterparts. Nevertheless, we believe that our continuous-time results will serve as a fundamental building block of any argument in that direction. Indeed, our diverse set of simulations, which are discrete-time implementations, are in perfect agreement with our theoretical predictions showcasing the applicability of our results.*

## A.2 Background in convex optimization

> For the sake of completeness, we recall here the definition of (strict) convex/concave function and its first order necessary and sufficient criterion. We will also discuss strong convexity and its second order characterizations.

We will be interested in notions from convex optimization throughout this work

**Definition 9** ([11, p. 67]). *Let $f : \mathbb{R}^n \to \mathbb{R}$ be a function then*

- *$f$ is convex if*
$$\forall \mathbf{x}, \mathbf{y} \in \mathbb{R}^n, t \in [0, 1] : f(t\mathbf{x} + (1-t)\mathbf{y}) \leq t f(\mathbf{x}) + (1-t) f(\mathbf{y})$$

- *$f$ is strictly convex if*
$$\forall \mathbf{x}, \mathbf{y} \in \mathbb{R}^n, t \in (0, 1) : f(t\mathbf{x} + (1-t)\mathbf{y}) < t f(\mathbf{x}) + (1-t) f(\mathbf{y})$$

- *$f$ is (strictly) concave if $-f$ is (strictly) convex.*

We will also use the first order characterizations of convex and concave functions

**Theorem 13** ([11, p. 69-70]). *Let $f : \mathbb{R}^n \to \mathbb{R}$ be a differentiable function.*

- *$f$ is convex if and only if $\forall \mathbf{x}, \mathbf{y} \in \mathbb{R}^n : f(\mathbf{y}) \geq f(\mathbf{x}) + \nabla f(\mathbf{x})^T (\mathbf{y} - \mathbf{x})$*
- *$f$ is concave if and only if $\forall \mathbf{x}, \mathbf{y} \in \mathbb{R}^n : f(\mathbf{y}) \leq f(\mathbf{x}) + \nabla f(\mathbf{x})^T (\mathbf{y} - \mathbf{x})$*

To establish convergence rates, we will use the notion of strong convexity

**Definition 10** ([49, p. 63]). *A continuously differentiable function $f$ of $\mathbb{R}^n$ will be called $\mu$ strongly convex for a positive constant $\mu$ if for all $\mathbf{x}, \mathbf{y} \in \mathbb{R}^n$ we have*

$$f(\mathbf{y}) \geq f(\mathbf{x}) + \langle \nabla f(\mathbf{x}), \mathbf{y} - \mathbf{x} \rangle + \frac{\mu}{2} \|\mathbf{x} - \mathbf{y}\|^2$$

We will also use second order characterizations of strong convexity

**Theorem 14** ([49, p. 65]). *A twice continuously differentiable function $f$ is $\mu$ strongly convex for a positive constant $\mu$ if and only if for all $\mathbf{x} \in \mathbb{R}^n$ we have*

$$\nabla^2 f(\mathbf{x}) \geq \mu I$$

Symmetrically, a function will be called $\mu$ strongly concave if $-f$ is $\mu$ strongly convex.

## A.3 Background in Game Theory

> In this short section, we remind to the reader a generalization of Von-Neumann's Minimax theorem, which we will exploit to analyze the equilibrium solution of the different GANs' architectures. A special case of Fan's minimax theorem is the following

**Corollary 2** (Fan's minimax theorem, [22]). *Let $X \subset \mathbb{R}^n$ and $Y \subset \mathbb{R}^m$ be convex non-empty sets. Suppose that $X$ is compact and $f : X \times Y \to \mathbb{R}$ is a function such that $f(\cdot, y)$ is lower semicontinuous on $X$ for each $y \in Y$ and that $f$ is convex concave. Then we have that*

$$\min_{x \in X} \sup_{y \in Y} f(x, y) = \sup_{y \in Y} \min_{x \in X} f(x, y).$$

# B Preliminaries

The below time-reparametrization lemma shows that the solution for a non-autonomous system, multiplicative to a gradient flow can be derived by just time-rescaling of the solution of the simplified gradient ascent dynamics. Indeed, since the multiplicative term is common across all terms of the vector field then over the time it dictates only the magnitude of the vector field (the speed of the motion), but does not affect the directionality other than moving backwards or forwards along the same trajectory.

**Lemma 1.** *Let $k : \mathbb{R}^d \to \mathbb{R}$ be a $C^2$ function. Let $h : \mathbb{R} \to \mathbb{R}$ be a $C^1$ function and $\mathbf{x}(t)$ denote the unique solution of the dynamical system $\Sigma_1$. Then the unique solution for dynamical system $\Sigma_2$ is $\mathbf{z}(t) = \mathbf{x}(\int_0^t h(s)\mathrm{d}s)$*

$$\left\{ \begin{matrix} \dot{\mathbf{x}} & = & \nabla k(\mathbf{x}) \\ \mathbf{x}(0) & = & \mathbf{x}_{init} \end{matrix} \right\} : \Sigma_1 \qquad \left\{ \begin{matrix} \dot{\mathbf{z}} & = & h(t)\nabla k(\mathbf{z}) \\ \mathbf{z}(0) & = & \mathbf{x}_{init} \end{matrix} \right\} : \Sigma_2 \tag{5}$$

*Proof.* Firstly, notice that it holds $\mathbf{x}(0) = \mathbf{x}_{\text{init}}$ and $\dot{\mathbf{x}} = \nabla k(\mathbf{x})$, since $\mathbf{x}$ is the unique solution of $\Sigma_1$. It is easy to check that:

$$\mathbf{z}(0) = \mathbf{x}\left(\int_0^0 h(s)\mathrm{d}s\right) = \mathbf{x}(0) = \mathbf{x}_{\text{init}}$$

$$\dot{\mathbf{z}} = \dot{\mathbf{x}}\left(\int_0^t h(s)\mathrm{d}s\right) \times \frac{\mathrm{d}[\int_0^t h(s)\mathrm{d}s]}{\mathrm{d}t}$$

$$= \nabla k\left(\mathbf{x}\left(\int_0^t h(s)\mathrm{d}s\right)\right) h(t) = \nabla k(\mathbf{z})h(t)$$

$\square$

**Remark 2.** *Notice that Lemma 1 applies* a-fortiori *in the class of HCCs games. Indeed, it is important to observe that allowing $h(t)$ to be any smooth function of time already covers already the cases where $h(t)$ depends also on $\mathbf{z}(t)$. The key intuition is that for the needs of our existential result, we can always assume that the exact solution of $\Sigma_2$ is known. Thus we can always substitute the $\mathbf{z}(t)$-dependent $h(t)$ with an expression that depends only directly on $t$.*

**Example 1.** *Let's consider the simplest example where $n = 1$ and $k(x) = x^2/2$ with for a state-dependent multiplicative factor $h(t) = z(t)$. Then,*

- $\Sigma_1 \equiv \{\dot{x}(t) = x(t), x(0) = 1\}$, *whose the unique solution is $x(t) = e^t$.*

- $\Sigma_2 \equiv \{\dot{z}(t) = h(t)z(t) = z(t)^2, z(0) = 1\}$ *whose unique solution is $z(t) = 1/(1-t)$.*

*We can still apply the aforementioned reparametrization lemma by substituting $h(t) = z(t)$ with $h(t) = 1/(1-t)$. In other words, any $h(x,t)$ can always be written given any initial condition as $h_x(t)$.*

**Remark 3.** *It is worth mentioning that due to the multiplicative factor $h(t)$ in $\Sigma_2$ and as a result the time reparamatrization factor $\int_0^t h(s)\mathrm{d}s$ can take both positive or negative values based on the GDA dynamics. By Lemma 1, this allows the GDA dynamics to visit states that are visited by the simple GD dynamics for $t < 0$.*

In order to leverage the convex-concave properties of the operators in our hidden structure under the Gradient Descent Ascent dynamics we need to recover the equivalent system $(T)$ in the operator space $\begin{pmatrix} \dot{\mathbf{F}} \\ \dot{\mathbf{G}} \end{pmatrix} = T\left[\begin{pmatrix} \mathbf{F} \\ \mathbf{G} \end{pmatrix}\right]$.

$$(\Sigma) := \begin{cases} \dot{\boldsymbol{\theta}} &=& - & \nabla L(\mathbf{F}(\boldsymbol{\theta}), \mathbf{G}(\boldsymbol{\phi})) \\ \dot{\boldsymbol{\phi}} &=& & \nabla L(\mathbf{F}(\boldsymbol{\theta}), \mathbf{G}(\boldsymbol{\phi})) \end{cases} \equiv \begin{cases} \dot{\boldsymbol{\theta}}_i &=& - & \nabla_{\boldsymbol{\theta}_i} f_i(\boldsymbol{\theta}_i)\partial L(\mathbf{F}(t), \mathbf{G}(t))/\partial f_i \\ \dot{\boldsymbol{\phi}}_j &=& & \nabla_{\boldsymbol{\phi}_j} g_j(\boldsymbol{\phi}_j)\partial L(\mathbf{F}(t), \mathbf{G}(t))/\partial g_j \end{cases}$$

From this point, applying the aforementioned lemma, under GDA each $f_i$ and $g_j$ follows a time dependent rescaling of the corresponding gradient ascent solution. Exploiting the monotonicity of $f_i(t)$ and $g_j(t)$ under gradient ascent, we can construct an invertible map between the parameter space $\{(\boldsymbol{\theta}_i, \boldsymbol{\phi}_j)\}$ and the operator space $\{(f_i, g_j)\}$ which allows us to construct the equivalent system $T$ in the operator space. *Notice that the properties of gradient ascent are crucial since the operator space can be arbitrarily smaller in dimension. In this case a smooth invertible map that is common for all initializations cannot exist.*

**Theorem 1.** *For each initialization $(\boldsymbol{\theta}(0), \boldsymbol{\phi}(0))$ of Equation (1), there are $C^1$ functions $X_{\boldsymbol{\theta}_i(0)}$, $X_{\boldsymbol{\phi}_j(0)}$ such that $\boldsymbol{\theta}_i(t) = X_{\boldsymbol{\theta}_i(0)}(f_i(t))$ and $\boldsymbol{\phi}_j(t) = X_{\boldsymbol{\phi}_j(0)}(g_j(t))$. If $(\boldsymbol{\theta}(t), \boldsymbol{\phi}(t))$ satisfy Equation (1) then $f_i(t) = f_i(\boldsymbol{\theta}_i(t))$ and $g_j(t) = g_j(\boldsymbol{\phi}_j(t))$ satisfy*

$$\dot{f}_i = -\|\nabla_{\boldsymbol{\theta}_i} f_i(X_{\boldsymbol{\theta}_i(0)}(f_i))\|^2 \frac{\partial L}{\partial f_i}(\mathbf{F}, \mathbf{G})$$

$$\dot{g}_j = \|\nabla_{\boldsymbol{\phi}_j} g_j(X_{\boldsymbol{\phi}_j(0)}(g_j))\|^2 \frac{\partial L}{\partial g_j}(\mathbf{F}, \mathbf{G})$$

(3)

*Proof.* Let us first study a simpler dynamical system $(\Sigma^*)$ with unique solution of $\gamma_{\boldsymbol{\theta}_i(0)}(t)$.

$$(\Sigma^*) \equiv \begin{cases} \dot{\mathbf{z}} &=& \nabla f_i(\mathbf{z}) \\ \mathbf{z}(0) &=& \boldsymbol{\theta}_i(0) \end{cases}$$

It is easy to observe that:

$$\dot{f}_i = \nabla f(\mathbf{z})\dot{\mathbf{z}} = \|\nabla f(\mathbf{z})\|^2$$

If $\boldsymbol{\theta}_i(0)$ is a stationary point of $f_i$ then the trajectory of $\mathbf{z}$ is a single point. But the trajectory of $\boldsymbol{\theta}_i$ under the dynamics of Equation (1) is also a single point so we can pick the following function

$$X_{\boldsymbol{\theta}_i(0)}(f_i) = \boldsymbol{\theta}_i(0).$$

On the other hand if $\boldsymbol{\theta}_i(0)$ is not a stationary point of $f_i$, $f_i$ continuously increases along the trajectory of $(\Sigma^*)$. Therefore $A_{\boldsymbol{\theta}_i(0)}(t) = f_i(\gamma_{\boldsymbol{\theta}_i(0)}(t))$ is an increasing function and therefore invertible. Let us call $A_{\boldsymbol{\theta}_i(0)}^{-1}(f_i)$ the inverse.

Let's recall now the $\boldsymbol{\theta}_i$ part of the dynamical system of interest Equation (1)

$$\dot{\boldsymbol{\theta}}_i = -\nabla_{\boldsymbol{\theta}_i} f_i(\boldsymbol{\theta}_i)\frac{\partial L}{\partial f_i}(\mathbf{F}(\boldsymbol{\theta}), \mathbf{G}(\boldsymbol{\phi}))$$

initialized at $\boldsymbol{\theta}_i(0)$. Applying Lemma 1 for the first equation with

$$h(t) = -\frac{\partial L}{\partial f_i}(\mathbf{F}(\boldsymbol{\theta}(t)), \mathbf{G}(\boldsymbol{\phi}(t)))$$

we have that under the dynamics of Equation (1)

$$\boldsymbol{\theta}_i(t) = \gamma_{\boldsymbol{\theta}_i(0)}\left(\int_0^t h(s)\mathrm{d}s\right)$$

(P)

Thus it holds

$$f_i(\boldsymbol{\theta}_i(t)) = f\left(\gamma_{\boldsymbol{\theta}_i(0)}\left(\int_0^t h(s)\mathrm{d}s\right)\right) = A_{\boldsymbol{\theta}_i(0)}\left(\int_0^t h(s)\mathrm{d}s\right)$$

or equivalently

$$\int_0^t h(s)\mathrm{d}s = A_{\boldsymbol{\theta}_i(0)}^{-1}(f_i(\boldsymbol{\theta}_i(t)))$$

Plugging in back to Equation (P)

$$\boldsymbol{\theta}_i(t) = \gamma_{\boldsymbol{\theta}_i(0)}(A^{-1}_{\boldsymbol{\theta}_i(0)}(f_i(\boldsymbol{\theta}_i(t))))$$

Therefore we can pick

$$X_{\boldsymbol{\theta}_i(0)}(f_i) = \gamma_{\boldsymbol{\theta}_i(0)}(A^{-1}_{\boldsymbol{\theta}_i(0)}(f_i))$$

which is $C^1$ as composition of $C^1$ functions. We can perform an equivalent analysis for $\boldsymbol{\phi}_j(0)$ and $g_j$ to pick $C^1$ function $X_{\boldsymbol{\phi}_j(0)}$. Let us now track the time derivative of $f_i(\boldsymbol{\theta}_i)$ and $g_j(\boldsymbol{\phi}_j)$

$$\dot{f}_i = \nabla_{\boldsymbol{\theta}_i} f_i(\boldsymbol{\theta}_i)\dot{\boldsymbol{\theta}}_i = \|\nabla_{\boldsymbol{\theta}_i} f_i(\boldsymbol{\theta}_i)\|^2 \frac{\partial L}{\partial f_i}(\mathbf{F}, \mathbf{G})$$

$$\dot{g}_j = \nabla_{\boldsymbol{\phi}_j} g_j(\boldsymbol{\phi}_j)\dot{\boldsymbol{\phi}}_j = \|\nabla_{\boldsymbol{\phi}_j} g_j(\boldsymbol{\phi}_j)\|^2 \frac{\partial L}{\partial g_j}(\mathbf{F}, \mathbf{G})$$

We can now replace $\boldsymbol{\theta}_i = X_{\boldsymbol{\theta}_i(0)}(f_i)$ and $\boldsymbol{\phi}_j = X_{\boldsymbol{\phi}_j(0)}(g_j)$ to get the equations required. $\qquad \square$

---

The XOR-functions in biological applications of Section 4 exemplify a set of cases where safety analysis is tractable. Much more generally, the separability assumption between the variables of $f_i$ and $g_j$ allows for a much more general positive result. Indeed, below we show that under standard assumptions on $f_i$ and $g_j$ and a target equilibrium $(\mathbf{p}, \mathbf{q})$ with

$$p_i \in (\max \text{LocalMin}(f_i), \min \text{LocalMax}(f_i))$$
$$q_j \in (\max \text{LocalMin}(g_j), \min \text{LocalMax}(g_j))$$

almost all initializations are safe by the Center-Stable Manifold-Theorem.

---

**Theorem 2.** *If $f_i$ and $g_j$ have isolated stationary points, only strict saddle points, compact sublevel-sets, both equilibria $p_i \in (\max \text{LocalMin}(f_i), \min \text{LocalMax}(f_i))$ and $q_j \in (\max \text{LocalMin}(g_j), \min \text{LocalMax}(g_j))$, then almost all initializations are safe for a $(\mathbf{p}, \mathbf{q}) \in Solution(L)$.*

*Proof.* Our proof structure goes as follows: Initially, we will prove that for each $i$ and for all but a measure zero sets of $\boldsymbol{\theta}_i(0)$ it holds that $p_i \in \text{Im}_{f_i}(\boldsymbol{\theta}_i(0))$ and $\boldsymbol{\theta}_i(0)$ is not a stationary point of $f_i$. Correspondingly, for each $j$ and for all but a measure zero sets of $\boldsymbol{\phi}_j(0)$, it holds that $q_j \in \text{Im}_{g_j}(\boldsymbol{\phi}_j(0))$ and $\boldsymbol{\phi}_j(0)$ is not a stationary points of $g_j$. Thus, since $(n, m)$ are finite, for all but a measure zero set of initializations $(\boldsymbol{\theta}(0), \boldsymbol{\phi}(0))$ are safe for $(\mathbf{p}, \mathbf{q})$.

What remains to prove is that for almost all $\boldsymbol{\theta}_i(0)$ it holds that $p_i \in \text{Im}_{f_i}(\boldsymbol{\theta}_i(0))$ and $\boldsymbol{\theta}_i(0)$ is not a stationary point of $f_i$. The proof for $g_j$ is completely symmetrical. Observe that the stationary points of $f_i$ are isolated. So we clearly have that almost all $\boldsymbol{\theta}_i(0) \in \mathbb{R}^{n_i}$ are not stationary points of $f_i$. We will then break the proof for $p_i \in \text{Im}_{f_i}(\boldsymbol{\theta}_i(0))$ in two pieces.

For the first piece we will prove that for almost all $\boldsymbol{\theta}_i(0)$ we have that $p_i < \sup \text{Im}_{f_i}(\boldsymbol{\theta}_i(0))$. To do this we need to study the dynamics of $\Sigma_1$ as $t \to \infty$. Let $\boldsymbol{\theta}_i(t)$ be the solution of $\Sigma_1$. As $t \to \infty$, either $\|\boldsymbol{\theta}_i(t)\|$ is bounded or it goes to $\infty$. For the case of $\|\boldsymbol{\theta}_i(t)\| \to \infty$ we know that $f_i(\boldsymbol{\theta}_i(t)) \to \infty$ since $f_i$ has compact sublevel sets and thus $p_i < \sup \text{Im}_{f_i}(\boldsymbol{\theta}_i(0))$ follows directly. For the remaining cases, we know that $\boldsymbol{\theta}_i(t)$ is bounded and thus it has a connected $\omega$ limit set. But since only stationary points of $f_i$ can be limit sets of $\boldsymbol{\theta}_i(t)$ and they are isolated, this means that $\boldsymbol{\theta}_i(t)$ has exactly one limit point and as a result it converges to this limit point as $t \to \infty$. Let us call this point $\mathbf{r}$. Since $f_i(\boldsymbol{\theta}_i(t))$ is increasing because of the gradient ascent dynamics of $\Sigma_1$, we have that $\sup \text{Im}_{f_i}(\boldsymbol{\theta}_i(0)) = f_i(\mathbf{r})$. Since $r$ is a stationary point of $f_i$, it is either either a local minimum, a local maximum or a saddle point of $f_i$. Clearly $\mathbf{r}$ cannot be a local minimum since $f_i(\boldsymbol{\theta}_i(t))$ is increasing so it cannot converge to a local minimum of $f_i$. Regarding saddle points, by assumption they are all strict and thus by the Stable Manifold Theorem (Theorem 12) only a zero measure set can converge to any of the isolated and thus countable saddle points. This leaves us with the case of $r$ being a local maximum in which case we have that $f(\mathbf{r}) \geq \min \text{LocalMax}(f_i)$. But by assumption $p_i < \min \text{LocalMax}(f_i) \leq f_i(\mathbf{r}) = \sup \text{Im}_{f_i}(\boldsymbol{\theta}_i(0))$.

For the second piece we will prove that for almost all $\boldsymbol{\theta}_i(0)$ we have that $p_i > \inf \operatorname{Im}_{f_i}(\boldsymbol{\theta}_i(0))$. To do this we need to study the dynamics of $\Sigma_1$ as $t \to -\infty$. Let $\boldsymbol{\theta}_i(t)$ be the solution of $\Sigma_1$. As $t \to -\infty$, either $\|\boldsymbol{\theta}_i(t)\|$ is bounded or it goes to $\infty$. For the case of $\|\boldsymbol{\theta}_i(t)\| \to \infty$ we know that $f_i(\boldsymbol{\theta}_i(t)) \to \infty$ since $f_i$ has compact sublevel sets. But this is a contradiction since $f_i(\boldsymbol{\theta}_i(t))$ is increasing so it cannot approach $\infty$ as $t \to -\infty$. For the remaining cases, just like above we have that $\boldsymbol{\theta}_i(t)$ converges to a point $\mathbf{r}$ and because of the gradient ascent dynamics of $\Sigma_1$, we have that $\inf \operatorname{Im}_{f_i}(\boldsymbol{\theta}_i(0)) = f_i(\mathbf{r})$. Since $r$ is a stationary point of $f_i$, it is either either a local minimum, a local maximum or a saddle point of $f_i$. Clearly $\mathbf{r}$ cannot be a local maximum since $f_i(\boldsymbol{\theta}_i(t))$ is increasing so it cannot converge to a local maximum of $f_i$ as $t \to -\infty$. Once again for saddle points, we know from (Theorem 12) only a zero measure set can converge to any of the isolated and thus countable saddle points. This leaves us with the case of $r$ being a local minimum in which case we have that $f(\mathbf{r}) \leq \max \operatorname{LocalMin}(f_i))$. But by assumption $p_i > \max \operatorname{LocalMin}(f_i)) \geq f_i(\mathbf{r}) = \inf \operatorname{Im}_{f_i}(\boldsymbol{\theta}_i(0))$. $\qquad\square$

**Remark 4.** *The key message of the above theorem is that the safety condition is a reasonable and relatively minimal assumption. Going beyond this type of result remains difficult as even in non-convex optimization, understanding which initializations lead to which local minimum of a loss function under gradient descent is a rather hard problem in its full generality, so no meaningful progress is possible without constraining ourselves in special cases with additional structure. Thus, instead of reiterating negative results based on unfortunate initializations that are not in agreement with the empirical success of GANs, we choose to study the dynamics of HCC games under safety.*

## C   Hidden Convex Concave Games

In this section, we analyze the derived stability properties of the hidden convex concave games. It is worth mentioning that without strict/strong convexity/concavity from at least one of the operators, the quality of the results are limited to "Lyapunov Stability". Firstly, we present a construction of a Lyapunov function for the operators' dynamics **Theorem 3**. Then, in **Theorem 4** and **Theorem 5** we explore the stability of the initial conditions in the parameter space.

### C.1   General case

The following theorem presents the construction of a Lyapunov potential function for the induced operator dynamics. To motivate its construction, we can study a fundamental convex-concave function $L(x, y) = (x - p)^2 - (y - q)^2$ with saddle point $(p, q)$. Under the gradient-descent-ascent dynamics

$$(T) := \begin{cases} \dot{x} & = & - & \nabla_x L(x, y) & \text{(minimization of convex part)} \\ \dot{y} & = & & \nabla_y L(x, y) & \text{(maximization of concave part)} \end{cases}.$$

it is easy to check that $H(x, y) = (x - p)^2 + (y - q)^2$ meets all the criteria of a Lyapunov function. The construction below extends this argument to any convex-concave function $L(\mathbf{F}, \mathbf{G})$ and bypasses the more complex multiplicative terms for the gradient induced dynamics of Theorem 1. *Notice that*

$$H(\mathbf{F}, \mathbf{G}) = \sum_{i=1}^{N} \int_{p_i}^{f_i} \frac{z - p_i}{\|\nabla f_i(X_{\boldsymbol{\theta}_i(0)}(z))\|^2} \mathrm{d}z + \sum_{j=1}^{M} \int_{q_j}^{g_j} \frac{z - q_j}{\|\nabla g_j(X_{\boldsymbol{\phi}_j(0)}(z))\|^2} \mathrm{d}z$$

*coincides with the $\ell_2^2$ distance from $(\mathbf{p}, \mathbf{q})$ in the case of gradient norms equal to one, i.e.*

$$\|\nabla f_i\|^2 = \|\nabla g_j\|^2 = 1$$

**Lemma 2.** *If $L$ is convex concave and $(\boldsymbol{\phi}(0), \boldsymbol{\theta}(0))$ is a safe for $(\mathbf{p}, \mathbf{q}) \in Solution(L)$, then the following quantity is non-increasing under the dynamics of Equation (3):*

$$H(\mathbf{F}, \mathbf{G}) = \sum_{i=1}^{N} \int_{p_i}^{f_i} \frac{z - p_i}{\|\nabla f_i(X_{\boldsymbol{\theta}_i(0)}(z))\|^2} \mathrm{d}z + \sum_{j=1}^{M} \int_{q_j}^{g_j} \frac{z - q_j}{\|\nabla g_j(X_{\boldsymbol{\phi}_j(0)}(z))\|^2} \mathrm{d}z \qquad (4)$$

*Proof.* Simple substitution gets us the following

$$\dot{H} = -\sum_{i=1}^{N}(f_i - p_i)\frac{\partial L}{\partial f_i}(\mathbf{F}, \mathbf{G}) + \sum_{j=1}^{M}(g_j - q_j)\frac{\partial L}{\partial g_j}(\mathbf{F}, \mathbf{G})$$

$$= -\langle \mathbf{F} - \mathbf{p}, \nabla_{\mathbf{F}}L(\mathbf{F}, \mathbf{G})\rangle + \langle \mathbf{G} - \mathbf{q}, \nabla_{\mathbf{G}}L(\mathbf{F}, \mathbf{G})\rangle$$

By Theorem 13 for the convex $L(\cdot, \mathbf{G})$ and concave $L(\mathbf{F}, \cdot)$.

$$-\langle \mathbf{F} - \mathbf{p}, \nabla_{\mathbf{F}}L(\mathbf{F}, \mathbf{G})\rangle \le L(\mathbf{p}, \mathbf{G}) - L(\mathbf{F}, \mathbf{G})$$
$$\langle \mathbf{G} - \mathbf{q}, \nabla_{\mathbf{G}}L(\mathbf{F}, \mathbf{G}) \le L(\mathbf{F}, \mathbf{G}) - L(\mathbf{F}, \mathbf{q})$$

Thus we can end up writing

$$\dot{H} \le L(\mathbf{p}, \mathbf{G}) - L(\mathbf{F}, \mathbf{G}) + L(\mathbf{F}, \mathbf{G}) - L(\mathbf{F}, \mathbf{q})$$
$$\le L(\mathbf{p}, \mathbf{G}) - L(\mathbf{p}, \mathbf{q}) + L(\mathbf{p}, \mathbf{q}) - L(\mathbf{F}, \mathbf{q}) \le 0$$

The last inequality holds since $(\mathbf{p}, \mathbf{q}) \in Solution(L)$. Indeed, if $(\mathbf{p}, \mathbf{q})$ is a saddle point of $L$ then $L(\mathbf{p}, \mathbf{G}) \le L(\mathbf{p}, \mathbf{q}) \le L(\mathbf{F}, \mathbf{q})$. $\qquad \square$

**Theorem 3.** *If $L$ is convex concave and $(\boldsymbol{\phi}(0), \boldsymbol{\theta}(0))$ is a safe for $(\mathbf{p}, \mathbf{q}) \in Solution(L)$, then $(\mathbf{p}, \mathbf{q})$ is stable for Equation (3).*

*Proof.* Leveraging Lemma 2, there is a function $H$ which is well defined in $D = \{\mathrm{Im}_{f_i}(\boldsymbol{\theta}_i(0))\}_{i=1}^{N} \times \{\mathrm{Im}_{g_j}(\boldsymbol{\phi}_j(0))\}_{j=1}^{M}$ and in this domain $\dot{H} \le 0$. Given the safety conditions we know that $(\mathbf{p}, \mathbf{q}) \in D$. Observe that for the proposed function, it holds that $H(\mathbf{p}, \mathbf{q}) = 0$. Also for each $f_i$ and $g_j$ term in $H$ we know that it has its minimum of value 0 at the corresponding $p_i$ and $q_j$. We can deduce this by taking the derivative of each term to study its monotonicity. For example, the $f_i$ terms are strictly increasing in $f_i > p_i$ and strictly decreasing in $f_i < p_i$. Thus for all $D - \{(\mathbf{p}, \mathbf{q})\}$, $H > 0$. Applying Theorem 9 for the continuously differentiable $H$ we have that $(\mathbf{p}, \mathbf{q})$ is stable for Equation (3). $\qquad \square$

> In the following example, we examine how it is possible to transfer the stability properties between two (topological conjugate) dynamical systems.

**Theorem 4.** *If $f_i$ and $g_j$ are sigmoid functions and $L$ is convex concave and there is a $(\boldsymbol{\phi}(0), \boldsymbol{\theta}(0))$ that is safe for $(\mathbf{p}, \mathbf{q}) \in Solution(L)$, then $(\mathbf{F}^{-1}(\mathbf{p}), \mathbf{G}^{-1}(\mathbf{q}))$ is stable for Equation (1).*

*Proof.* Firstly, we recall the property of sigmoid's gradient:

$$\frac{\mathrm{d}\sigma(x)}{\mathrm{d}x} = \sigma(x)(1 - \sigma(x)).$$

Thus the transformed dynamical system in the operator space can be written as:

$$(T) := \begin{cases} \dot{f}_i &= & - & f_i^2(1 - f_i)^2 \frac{\partial L}{\partial f_i}(\mathbf{F}, \mathbf{G}) \\ \dot{g}_j &= & & g_j^2(1 - g_j)^2 \frac{\partial L}{\partial g_j}(\mathbf{F}, \mathbf{G}) \end{cases}$$

*Notice that*

1. *The dynamical system $(T)$ in the operator space is independent of the initial conditions. In fact, the dynamical system of $(T)$ and the one of Equation (1), called $(\Sigma)$ for short, are diffeomorphic for all initializations, not just a specific trajectory.*

2. *Since $(\boldsymbol{\theta}(0), \boldsymbol{\phi}(0))$ is safe, using Theorem 3 we get that $(\mathbf{p}, \mathbf{q})$ is stable for $(T)$.*

We would like to prove that for every open neighborhood $V$ of $(\mathbf{F}^{-1}(\mathbf{p}), \mathbf{G}^{-1}(\mathbf{q}))$ there exists an open neighborhood $U$ of $(\mathbf{F}^{-1}(\mathbf{p}), \mathbf{G}^{-1}(\mathbf{q}))$ such that

$$(\boldsymbol{\theta}_{\text{init}}, \boldsymbol{\phi}_{\text{init}}) \in U \implies \forall t \geq 0 : (\boldsymbol{\theta}(t), \boldsymbol{\phi}(t)) \in V.$$

Using the diffeomorphism $\gamma = \gamma_{\Sigma \to T}$ between GDA dynamics of $(\Sigma)$ and $(T)$, $\gamma(V)$ is an open neighborhood of $(\mathbf{p}, \mathbf{q})$ since $V$ is open and $\gamma((\mathbf{F}^{-1}(\mathbf{p}), \mathbf{G}^{-1}(\mathbf{q}))) \equiv (\mathbf{p}, \mathbf{q}) \in \gamma(V)$. By Item 2, since $(\mathbf{p}, \mathbf{q})$ is stable for $(T)$ there is an open neighborhood $\tilde{U}$ of $(\mathbf{p}, \mathbf{q})$ such that:

$$(\mathbf{F}_{\text{init}}, \mathbf{G}_{\text{init}}) \in \tilde{U} \implies \forall t \geq 0 : (\mathbf{F}(t), \mathbf{G}(t)) \in \gamma(V)$$

or equivalently

$$\gamma(\boldsymbol{\theta}_{\text{init}}, \boldsymbol{\phi}_{\text{init}}) \in \tilde{U} \implies \forall t \geq 0 : \gamma(\boldsymbol{\theta}(t), \boldsymbol{\phi}(t)) \in \gamma(V)$$

Indeed, using the inverse diffeomorphism $\gamma^{-1}$, we can establish that for $U = \gamma^{-1}(\tilde{U})$ it holds that

$$(\boldsymbol{\theta}_{\text{init}}, \boldsymbol{\phi}_{\text{init}}) \in U \implies \forall t \geq 0 : (\boldsymbol{\theta}(t), \boldsymbol{\phi}(t)) \in V$$

$\square$

---

Until now, we have established the stability of a pair $(\mathbf{p}, \mathbf{q})$ for the induced dynamics $(T)$. By the construction of the induced dynamics, $(T)$ is coupled only with a very specific initial condition $(\boldsymbol{\theta}_{\text{init}}, \boldsymbol{\phi}_{\text{init}})$. In order to tackle the challenge of a stability result for a whole region of initial conditions, in the following lemma we prove that $r(\boldsymbol{\theta}, \boldsymbol{\phi}) = \|\mathbf{F}(\boldsymbol{\theta}) - \mathbf{p}\|^2 + \|\mathbf{G}(\boldsymbol{\phi}) - \mathbf{q}\|^2$ can work like an intrinsic measure of closeness for the $\{\boldsymbol{\theta}, \boldsymbol{\phi}\}$-parameter space around a hidden fixed point of the $\{\mathbf{F}, \mathbf{G}\}$-operator space. Under this "hidden" neighborhood notion, stability property can be taken by assuming the properness of the hidden operators.

---

**Theorem 5.** *Let $(\mathbf{p}, \mathbf{q}) \in Solution(L)$. Let $R_{f_i}$ and $R_{g_j}$ be the set of regular values[3] of $f_i$ and $g_j$ respectively. Assume that there is a $\xi > 0$ such that $[p_i - \xi, p_i + \xi] \subseteq R_{f_i}$ and $[q_j - \xi, q_j + \xi] \subseteq R_{g_j}$. Define*

$$r(t) = \|\mathbf{F}(\boldsymbol{\theta}(t)) - \mathbf{p}\|^2 + \|\mathbf{G}(\boldsymbol{\phi}(t)) - \mathbf{q}\|^2.$$

*If $f_i$ and $g_j$ are proper functions[4], then for every $\epsilon > 0$, there is an $\delta > 0$ such that*

$$r(0) < \delta \implies \forall t \geq 0 : r(t) < \epsilon.$$

*Proof.* Let us define the following sets

$$\begin{array}{llll}
\forall i \in [n] & : A_i = & \{ \boldsymbol{\theta}_i \in \mathbb{R}^{n_i} & | & f_i(\boldsymbol{\theta}_i) \in [p_i - \xi, p_i + \xi] \} \\
\forall j \in [m] & : B_j = & \{ \boldsymbol{\phi}_j \in \mathbb{R}^{m_j} & | & g_j(\boldsymbol{\phi}_j) \in [q_j - \xi, q_j + \xi] \}
\end{array}$$

Since $f_i$ and $g_j$ are proper $A_i$ and $B_j$ are compact sets. Thus, the continuous functions $\|\nabla f_i(\boldsymbol{\theta}_i)\|^2$ and $\|\nabla g_j(\boldsymbol{\phi}_j)\|^2$ have a minimum and maximum value on $A_i$ and $B_j$ respectively. Let us call $K_{f_i}$ and $K_{g_j}$ the maxima and $\kappa_{f_i}$ and $\kappa_{g_j}$ the minima. Observe that the minima and maxima must be all greater than zero since $[p_i - \xi, p_i + \xi]$ and $[q_j - \xi, q_j + \xi]$ are regular values. Let us define

$$\kappa = \min\{ \min_{1 \leq i \leq n} \kappa_{f_i}, \min_{1 \leq j \leq m} \kappa_{g_j} \}$$

$$K = \max\{ \max_{1 \leq i \leq n} K_{f_i}, \max_{1 \leq j \leq m} K_{g_j} \}$$

where $K \geq \kappa > 0$ as we discussed. Let us create the following set

$$S = \{(\boldsymbol{\theta}, \boldsymbol{\phi}) \in \mathbb{R}^N \times \mathbb{R}^M \mid \forall i \in [n] : \boldsymbol{\theta}_i \in A_i, \quad \forall j \in [m] : \boldsymbol{\phi}_j \in B_j\}$$

We can prove that every $(\boldsymbol{\theta}, \boldsymbol{\phi}) \in S$ is a safe initialization for $(\mathbf{p}, \mathbf{q})$. Of course, every $\boldsymbol{\theta}_i$ and $\boldsymbol{\phi}_j$ are not stationary points of $f_i$ and $g_j$ respectively. We also need to prove that the equilibrium $(\mathbf{p}, \mathbf{q})$ is feasible. We will prove this by contradiction. Let there be a $(\boldsymbol{\theta}, \boldsymbol{\phi}) \in S$ such that $(\mathbf{p}, \mathbf{q})$ is not feasible. Without loss of generality we can assume that there is an $i \in [n]$ such that $p_i \notin \text{Im}_{f_i}(\boldsymbol{\theta}_i)$.

---

[3]A value $a \in \text{Im} f$ is called a regular value of $f$ if $\forall q \in \text{dom} f : f(q) = a$, it holds $\nabla f(q) \neq \mathbf{0}$.

[4]A function is proper if inverse images of compact subsets are compact.

The case for the $g_j$ is symmetrical. Along the gradient ascent trajectory of $f_i$ with initialization at $\boldsymbol{\theta}_i$, observe that $f_i(t)$ cannot attain an infimum or a supremum in $[p_i - \xi, p_i + \xi]$ because there are no stationary points of $f_i$ in $A_i$. Observe also that at initialization $f_i(\boldsymbol{\theta}_i) \in [p_i - \xi, p_i + \xi]$. Thus $[p_i - \xi, p_i + \xi] \subseteq \mathrm{Im}_{f_i}(\boldsymbol{\theta}_i)$, a contradiction.

Let us pick an initialization $(\boldsymbol{\theta}(0), \boldsymbol{\phi}(0))$ such that $r(0) \leq \xi^2$. It is clear that $(\boldsymbol{\theta}(0), \boldsymbol{\phi}(0)) \in S$ and so it is safe for $(\mathbf{p}, \mathbf{q})$. We can do the same steps as in Theorem 3 to prove that the function $H(\mathbf{F}, \mathbf{G})$ below does not increase under the dynamics of Equation (1):

$$H(\mathbf{F}, \mathbf{G}) = \sum_{i=1}^{N} \int_{p_i}^{f_i} \frac{z - p_i}{\|\nabla f_i(X_{\boldsymbol{\theta}_i(0)}(z))\|^2} \mathrm{d}z + \sum_{j=1}^{M} \int_{q_j}^{g_j} \frac{z - q_j}{\|\nabla g_j(X_{\boldsymbol{\phi}_j(0)}(z))\|^2} \mathrm{d}z$$

Observe that since $(\boldsymbol{\theta}(0), \boldsymbol{\phi}(0)) \in S$ we have that the interval between $p_i$ and $f_i(\boldsymbol{\theta}_i(0))$ belongs in $[p_i - \xi, p_i + \xi]$ and $\|\nabla f_i(\cdot)\|^2 \geq \kappa$ in this interval. Thus we can write

$$\frac{(f_i(\boldsymbol{\theta}_i(0)) - p_i)^2}{2\kappa} \geq \int_{p_i}^{f_i(\boldsymbol{\theta}_i(0))} \frac{z - p_i}{\|\nabla f_i(X_{\boldsymbol{\theta}_i(0)}(z))\|^2} \mathrm{d}z$$

Repeating the same argument for all $f_i$ and $g_j$ we have that

$$\frac{r(0)}{2\kappa} \geq H(\mathbf{F}(\boldsymbol{\theta}(0)), \mathbf{G}(\boldsymbol{\phi}(0))) \geq H(\mathbf{F}(\boldsymbol{\theta}(t)), \mathbf{G}(\boldsymbol{\phi}(t)))$$

Let us pick $r(0) < \min\{\xi^2, \xi^2 \frac{\kappa}{K}\} = \xi^2 \frac{\kappa}{K}$. We already know that trajectories start in $S$. We will prove that they also remain in $S$. We will do this by contradiction. If a trajectory escaped $S$, then without loss of generality this means that there is at least an $i \in [n]$ such that at some $t > 0$, $f_i(\boldsymbol{\theta}_i(t)) \notin [p_i - \xi, p_i + \xi]$. The case of $g_j$ is similar. Clearly we have that

$$\int_{p_i}^{f_i(\boldsymbol{\theta}_i(t))} \frac{z - p_i}{\|\nabla f_i(X_{\boldsymbol{\theta}_i(0)}(z))\|^2} \mathrm{d}z \geq \min\left\{ \int_{p_i}^{p_i - \xi} \frac{z - p_i}{\|\nabla f_i(X_{\boldsymbol{\theta}_i(0)}(z))\|^2} \mathrm{d}z, \int_{p_i}^{p_i + \xi} \frac{z - p_i}{\|\nabla f_i(X_{\boldsymbol{\theta}_i(0)}(z))\|^2} \mathrm{d}z \right\}$$

As above, we have that the gradients in the integrals of the right hand side are less or equal than $K$ so

$$\int_{p_i}^{f_i(\boldsymbol{\theta}_i(t))} \frac{z - p_i}{\|\nabla f_i(X_{\boldsymbol{\theta}_i(0)}(z))\|^2} \mathrm{d}z \geq \frac{\xi^2}{2K}.$$

The terms of $H$ are all non-negative so we have that

$$\frac{r(0)}{2\kappa} \geq H(\mathbf{F}(\boldsymbol{\theta}(t)), \mathbf{G}(\boldsymbol{\phi}(t))) \geq \int_{p_i}^{f_i(\boldsymbol{\theta}_i(t))} \frac{z - p_i}{\|\nabla f_i(X_{\boldsymbol{\theta}_i(0)}(z))\|^2} \mathrm{d}z \geq \frac{\xi^2}{2K}.$$

But $r(0) < \xi^2 \frac{\kappa}{K}$, a contradiction. So the trajectories will stay in $S$. We can then write

$$\int_{p_i}^{f_i(\boldsymbol{\theta}_i(t))} \frac{z - p_i}{\|\nabla f_i(X_{\boldsymbol{\theta}_i(0)}(z))\|^2} \mathrm{d}z \geq \frac{(f_i(\boldsymbol{\theta}_i(t)) - p_i)^2}{2K}.$$

Repeating the same argument for all $f_i$ and $g_j$ we have that

$$\frac{r(0)}{2\kappa} \geq H(\mathbf{F}(\boldsymbol{\theta}(t)), \mathbf{G}(\boldsymbol{\phi}(t))) \geq \frac{r(t)}{2K}.$$

For every $\epsilon > 0$, there is a positive $\delta = \frac{\min\{\xi^2, \epsilon\}\kappa}{K}$ such that
$$r(0) < \delta \implies r(t) < \epsilon.$$

$\square$

A special case of the above result is the standard convex-concave games:

**Corollary 3.** *Let $L(\mathbf{x}, \mathbf{y})$ be strictly convex concave and $Solution(L)$ is the non empty set of equilbria of $L$. Then $Solution(L)$ is locally asymptotically stable for continuous GDA dynamics.*

*Proof.* The proof of the above classical result can be derived by the straightforward application of Lemma 3 for the case of $\mathbf{F}(\mathbf{x}) = \mathbf{x}$ and $\mathbf{G}(\mathbf{y}) = \mathbf{y}$. Notice that *i)* if $\mathbf{F}, \mathbf{G}$ are the identity maps all the initial configurations are safe and *ii)* if $\|\nabla \mathbf{F}\|^2 = \|\nabla \mathbf{G}\|^2 = 1$, then the initialization-dependent Lyapunov functions coincide to a single Lyapunov function, which is actually the squared Euclidean distance $r(\boldsymbol{\theta}, \boldsymbol{\phi}) = \|\mathbf{F}(\boldsymbol{\theta}) - \mathbf{p}\|^2 + \|\mathbf{G}(\boldsymbol{\phi}) - \mathbf{q}\|^2 = \|\boldsymbol{\theta} - \mathbf{p}\|^2 + \|\boldsymbol{\phi} - \mathbf{q}\|^2$. $\square$

## C.2 Hidden strictly convex concave games

### C.2.1 Gradient Descent-Ascent Dynamics

> In the following preliminary result, we show that strict convexity or concavity in $L(\cdot, \cdot)$, for at least one of its arguments, suffices to yield locally asymptotic stability starting from a safe initial condition. Our argumentation leverages the power of Theorem 10 and combines the previous section stability results. Here, we will firstly outline the basic steps below:
>
> 1. We start by showing that there exists a compact set $\Omega \subset D$.
>
> 2. Therefore, since $\dot{H} \leq 0$ (Lyapunov property), any configuration $(\mathbf{F}(0), \mathbf{G}(0))$ starting from a bounded sub-level set $\Omega$ of $H$, will remain inside $\Omega$ over all time.
>
> 3. The second crucial observation is that thanks to the strictness on convexity or concavity of $L$, the largest invariant set of $\dot{H} = 0$ contains only points belonging to Von Neumann's Solution($L$).
>
> Then Theorem 10 implies the local asymptotic stability of set $Z$ for Equation (3).

**Lemma 3.** *Let $L$ be strictly convex concave and $Z \subset$ Solution($L$) is the non empty set of equilbria of $L$ for which $(\boldsymbol{\theta}(0), \boldsymbol{\phi}(0))$ is safe. Then $Z$ is locally asymptotically stable for Equation* (3).

*Proof.* Pick a point $(\mathbf{p}, \mathbf{q}) \in Z$. Since our initialization is safe for this saddle point, we can construct the $H$ function as in Theorem 3 and prove that it has the following property

$$\dot{H} \leq 0 \text{ in } D = \{\mathrm{Im}_{f_i}(\boldsymbol{\theta}_i(0))\}_{i=1}^{N} \times \{\mathrm{Im}_{g_j}(\boldsymbol{\phi}_j(0))\}_{j=1}^{M}$$

If $(\mathbf{F}(\boldsymbol{\theta}(0)), \mathbf{G}(\boldsymbol{\phi}(0))) = (\mathbf{p}, \mathbf{q})$ then the theorem holds trivially. Otherwise, take a ball $B$ centered at the equilibirum with a small enough radius such that it is contained in the interior of $D$.

$$H_0 = \min_{(\mathbf{F}, \mathbf{G}) \in \partial B} H(\mathbf{F}, \mathbf{G})$$

$$\Omega = \{(\mathbf{F}, \mathbf{G}) \in B | H(\mathbf{F}, \mathbf{G}) \leq H_0/2\}$$

We know that in both of the cases $H_0 > 0$ from Theorem 3.

Since $\dot{H} \leq 0$, starting in $\Omega$, it implies that $H(\mathbf{F}(t), \mathbf{G}(t)) \leq H_0$ for $t \geq 0$, so $\Omega$ is forward invariant. Since $\Omega \subset D$ we know that it is bounded. $\Omega$ is closed since it is a sublevel set of a continuous function. Notice that the restriction of $\Omega$ on $B$ does not affect the above properties since $\Omega$ is in the interior of $B$. Thus $\Omega$ is a compact forward invariant set, satisfying the requirement of Theorem 10

Let $E = \{(\mathbf{F}, \mathbf{G}) \in B | \dot{H}(\mathbf{F}, \mathbf{G}) = 0\}$. Without loss of generality we can assume that $L(\cdot, \mathbf{q})$ is strictly convex as the case of $L(\mathbf{p}, \cdot)$ being strictly concave is similar. In the following inequality

$$\dot{H} \leq L(\mathbf{p}, \mathbf{G}) - L(\mathbf{p}, \mathbf{q}) + L(\mathbf{p}, \mathbf{q}) - L(\mathbf{F}, \mathbf{q}) \leq 0$$

we know that $L(\mathbf{p}, \mathbf{G}) - L(\mathbf{p}, \mathbf{q}) \leq 0$ and $L(\mathbf{p}, \mathbf{q}) - L(\mathbf{F}, \mathbf{q}) \leq 0$.

So $\dot{H} = 0$ implies $L(\mathbf{p}, \mathbf{G}) = L(\mathbf{p}, \mathbf{q}) = L(\mathbf{F}, \mathbf{q})$. By the strict convexity of $L(\cdot, \mathbf{q})$ we know that this means that $\mathbf{F} = \mathbf{p}$. Let $\mathcal{M}$ be the largest invariant set inside $E$. By the properties of $\mathcal{M}$ being invariant subset of $E$ we have

$$(\mathbf{F}(0), \mathbf{G}(0)) \in \mathcal{M} \implies \forall t : \mathbf{F}(t) = \mathbf{p} \text{ and } L(\mathbf{p}, \mathbf{G}(t)) = L(\mathbf{p}, \mathbf{q})$$

Taking the time derivatives on each of the constant quantities, they should be zero.

$$\dot{f}_i = 0 \Rightarrow \qquad \forall i \in [N] : \quad \|\nabla_{\boldsymbol{\theta}_i} f_i(X_{\boldsymbol{\theta}_i(0)}(p_i))\|^2 \frac{\partial L}{\partial f_i}(\mathbf{p}, \mathbf{G}) = 0$$

$$L(\mathbf{p}, \dot{\mathbf{G}}(t)) = 0 \Rightarrow \qquad \sum_{j=1}^{M} \|\nabla_{\boldsymbol{\phi}_j} g_j(X_{\boldsymbol{\phi}_j(0)}(g_j))\|^2 \left[\frac{\partial L}{\partial g_j}(\mathbf{p}, \mathbf{G})\right]^2 = 0$$

We know that $\|\nabla_{\boldsymbol{\theta}_i} f_i(X_{\boldsymbol{\theta}_i(0)}(p_i))\| \neq 0$ by the safety conditions and that $\|\nabla_{\boldsymbol{\phi}_j} g_j(X_{\boldsymbol{\phi}_j(0)}(g_j))\|^2 \neq 0$ inside $D$ again by safety conditions. This implies

$$\forall i \in [N] : \frac{\partial L}{\partial f_i}(\mathbf{p}, \mathbf{G}) = 0$$

$$\forall j \in [M] : \frac{\partial L}{\partial g_j}(\mathbf{p}, \mathbf{G}) = 0$$

Thus $\mathcal{M}$ contains only stationary points of $L$ so $\mathcal{M} \subseteq \text{Solution}(L)$. In addition $\mathcal{M} \subseteq D$ so only stationary points of $L$ for which the initialization is safe are allowed so $\mathcal{M} \subseteq Z$. Applying Theorem 10 we have that for any initialization of Equation (3) inside $\Omega$, as $t \to \infty$ $(\mathbf{F}(t), \mathbf{G}(t))$ approaches $\mathcal{M}$ and thus $Z$ is locally asymptotically stable for Equation (3). $\square$

A special case of the above result is the standard convex-concave games:

**Corollary 4.** *Let $L(\mathbf{x}, \mathbf{y})$ be strictly convex concave and $\text{Solution}(L)$ is the non empty set of equilbria of L. Then $\text{Solution}(L)$ is locally asymptotically stable for continuous GDA dynamics.*

In the following main result of our work, we show that strict convexity or concavity in $L(\cdot, \cdot)$, for at least one of its arguments, suffices to yield a convergence result to a Von Neumann's $\text{Solution}(L)$ starting from a safe initial condition. In order to get convergence results for any safe initialization, we need to study the region of attraction of the set $Z \subset \text{Solution}(L)$. We refine the estimation of the region of attraction as proposed in Lemma 3 by analyzing the behavior of the level sets of $H$. More precisely, we show that the proposed Lyapunov function

$$H(\mathbf{F}, \mathbf{G}) = \sum_{i=1}^{N} \int_{p_i}^{f_i} \frac{z - p_i}{\|\nabla f_i(X_{\boldsymbol{\theta}_i(0)}(z))\|^2} dz + \sum_{j=1}^{M} \int_{q_j}^{g_j} \frac{z - q_j}{\|\nabla g_j(X_{\boldsymbol{\phi}_j(0)}(z))\|^2} dz$$

is radially unbounded. In other words, while the operators converges to their limit values (supremum/infimum of their domain) $H \to +\infty$. In order to show that we analyze the asymptotic behavior of $\int_c^{\mathcal{F}} \frac{1}{\|\nabla f_i\|^2}$, while $\mathcal{F} \to \sup f_i$. Hence,

A) Theorem 10 implies that the trajectory will approach the set of stationary points of $H$ or equivalently a set of Von Neumann's $\text{Solution}(L)$.

B) The stability of $\text{Solution}(L)$ and Theorem 11, leads to the conclusion that the trajectory will converges to a specific point of $\text{Solution}(L)$.

**Theorem 6.** *Let $L$ be strictly convex concave and $Z \subset \text{Solution}(L)$ is the non empty set of equilbria of L for which $(\boldsymbol{\theta}(0), \boldsymbol{\phi}(0))$ is safe. Under the dynamics of Equation (1) $(\mathbf{F}(\boldsymbol{\theta}(t)), \mathbf{G}(\boldsymbol{\theta}(t)))$ converges to a point in Z as $t \to \infty$.*

*Proof.* Again let's pick a point $(\mathbf{p}, \mathbf{q}) \in Z$. Since our initialization is safe for this saddle point, we can construct the $H$ function as in Theorem 3 and prove that it has the following property

$$\dot{H} \leq 0 \text{ in } D = \{\text{Im}_{f_i}(\boldsymbol{\theta}_i(0))\}_{i=1}^{N} \times \{\text{Im}_{g_j}(\boldsymbol{\phi}_j(0))\}_{j=1}^{M}$$

If $(\mathbf{F}(\boldsymbol{\theta}(0)), \mathbf{G}(\boldsymbol{\phi}(0))) = (\mathbf{p}, \mathbf{q})$ then the theorem holds trivially. Otherwise define

$$H_0 = H(\mathbf{F}(\boldsymbol{\theta}(0)), \mathbf{G}(\boldsymbol{\phi}(0)))$$
$$\Omega = \{(\mathbf{F}, \mathbf{G}) \in D | H(\mathbf{F}, \mathbf{G}) \leq H_0\}$$

where we know that $H_0 > 0$ from Theorem 3. Let us assume that indeed $\Omega$ is in the interior of $D$. Then, applying the same argumentation as in Lemma 3 combined with Theorem 3, all fixed points in $Z$ are stable. So applying Theorem 11 we get that the trajectory initialized at $(\mathbf{F}(\boldsymbol{\theta}(0)), \mathbf{G}(\boldsymbol{\phi}(0))) \in \Omega$ converges to a point in $Z$. It remains to prove our assertion about the set $\Omega$:

**Claim 1.** $\Omega$ *is in the interior of $D$.*

*Proof.* We will argue that as $(\mathbf{F}, \mathbf{G})$ approaches the boundary of $D$, the value of $H$ should become unbounded. If this is true then for the finite upper bound of $H_0$, $\Omega$ should have no points close to the boundary of $H$ and thus it should be in the interior.

As $(\mathbf{F}, \mathbf{G})$ approach the boundary of $D$, at least one of the variables $f_i$ or $g_j$ approaches the endpoints points of $\mathrm{Im}_{f_i}(\boldsymbol{\theta}_i(0))$ or $\mathrm{Im}_{g_j}(\boldsymbol{\phi}_j(0))$ respectively. We will study the case of $f_i$ since the case of $g_j$ is symmetrical. The endpoint $f_{is}$ can be either the supremum or the infimum of the gradient ascent trajectory on $f_i$ or $\pm\infty$ if they do not exist. Let $f_{is}$ be the supremum or $\infty$ depending on if the former exists. We can take the gradient ascent dynamics and apply Lemma 1 to get

$$\dot{f}_i = \|\nabla_{\boldsymbol{\theta}_i} f_i(X_{\boldsymbol{\theta}_i(0)}(f_i))\|^2$$

We know that $f_i(\boldsymbol{\theta}_i(t))$ goes to $f_{is}$ when initialized at $f_i(\boldsymbol{\theta}_i(0))$. Let us define the following function

$$a(f_i) = \int_{p_i}^{f_i} \frac{1}{\|\nabla f_i(X_{\boldsymbol{\theta}_i(0)}(z))\|^2} \, \mathrm{d}z$$

Observe that $\dot{a} = 1$, thus $\lim_{t \to \infty} a(f_i(t)) = \infty$. In other words

$$\lim_{t \to \infty} \int_{p_i}^{f_i(t)} \frac{1}{\|\nabla f_i(X_{\boldsymbol{\theta}_i(0)}(z))\|^2} \, \mathrm{d}z = \int_{p_i}^{f_{is}} \frac{1}{\|\nabla f_i(X_{\boldsymbol{\theta}_i(0)}(z))\|^2} \, \mathrm{d}z = \infty$$

Symmetrically if $f_{is}$ is the infimum or $-\infty$, then the limit above would be $-\infty$. In either case

$$f_i \to f_{is} \implies \int_{p_i}^{f_i} \frac{z - p_i}{\|\nabla f_i(X_{\boldsymbol{\theta}_i(0)}(z))\|^2} \, \mathrm{d}z \to \infty$$

For the last step it is important to note that $p_i$ is not at the boundary of $D$ based on the safety conditions. Therefore as $(\mathbf{F}, \mathbf{G})$ approach the boundary of $D$ in the dynamics of Equation (3), at least one of the terms of $H$ goes to infinity. Also note that all the terms of $H$ are individually non-negative so no matter what the other variables in $(\mathbf{F}, \mathbf{G})$ are doing they cannot stop $H \to \infty$. $\square$

$\square$

Again, a special case of the above result is the standard convex-concave games:

**Corollary 5.** *Let $L(\boldsymbol{x}, \boldsymbol{y})$ be strictly convex concave and $\mathrm{Solution}(L)$ is the non empty set of equilbria of $L$. Under the continuous GDA dynamics $(\boldsymbol{x}(t), \boldsymbol{y}(t))$ converges to a point in $\mathrm{Solution}(L)$ as $t \to \infty$.*

### C.2.2 Connections to Hamiltonian Descent

In GANs numerous learning heuristics are being tested and explored. One technique that has particular interesting theoretical justification as well as practical performance is Hamiltonian Gradient Descent (HGD). Understanding the convergence guarantees for HGD is an open research question [39, 8, 51]. We provide some new justification about its success in GANs by provably establishing convergence of a modified version of HGD in a relatively simple but illustrative subclass of hidden convex concave games, namely 2x2 hidden bi-linear games. This class of games is fairly expressive. Despite the restriction of planar bi-linear competition in the output space, the hidden game can have an arbitrary number of variables in the parameter space. It's important to note that given the bi-linear nature of competition, the classical GDA dynamics cycles instead of converging to the equilibrium as shown in [65]

More precisely, in the hidden 2x2 bi-linear game presented in [65], we have two functions $f : \mathbb{R}^N \to [0,1]$ and $g : \mathbb{R}^M \to [0,1]$ and two constants $(p, q) \in (0,1)^2$ where $(p, q)$ is the fully mixed equilibrium of the bi-linear game. Without loss of generality, we are interested in solving the following problem

$$\min_{\boldsymbol{\theta} \in \mathbb{R}^M} \max_{\boldsymbol{\phi} \in \mathbb{R}^N} (f(\boldsymbol{\theta}) - p)(g(\boldsymbol{\phi}) - q)$$

Defining $L(\boldsymbol{\theta}, \boldsymbol{\phi}) = (f(\boldsymbol{\theta}) - p)(g(\boldsymbol{\phi}) - q)$, the dynamics of HGD are:

$$\begin{aligned}
\dot{\boldsymbol{\theta}} &= -\frac{1}{2} \nabla_{\boldsymbol{\theta}} \|\nabla_{\boldsymbol{\phi}} L(\boldsymbol{\theta}, \boldsymbol{\phi})\|^2 - \frac{1}{2} \nabla_{\boldsymbol{\theta}} \|\nabla_{\boldsymbol{\theta}} L(\boldsymbol{\theta}, \boldsymbol{\phi})\|^2 \\
\dot{\boldsymbol{\phi}} &= -\frac{1}{2} \nabla_{\boldsymbol{\phi}} \|\nabla_{\boldsymbol{\theta}} L(\boldsymbol{\theta}, \boldsymbol{\phi})\|^2 - \frac{1}{2} \nabla_{\boldsymbol{\phi}} \|\nabla_{\boldsymbol{\phi}} L(\boldsymbol{\theta}, \boldsymbol{\phi})\|^2
\end{aligned} \tag{6}$$

Observe that the second term of each right hand side would be zero in a classical bi-linear game but involves second order derivatives of $f$ and $g$ in the case of hidden bi-linear games. To circumvent the complexities of the second order derivatives and mimic the classical bi-linear game we will study a modified version of Equation (6), namely:

$$\dot{\boldsymbol{\theta}} = -\frac{1}{2}\nabla_{\boldsymbol{\theta}}\|\nabla_{\boldsymbol{\phi}}L(\boldsymbol{\theta},\boldsymbol{\phi})\|^2 \qquad \dot{\boldsymbol{\phi}} = -\frac{1}{2}\nabla_{\boldsymbol{\phi}}\|\nabla_{\boldsymbol{\theta}}L(\boldsymbol{\theta},\boldsymbol{\phi})\|^2 \qquad (7)$$

Employing an analysis similar to the one in Section 3.2, we get the following convergence result:

**Theorem 15.** *Let* $(\boldsymbol{\theta}(0),\boldsymbol{\phi}(0))$ *be safe for* $(p,q)$. *Then* $(f(\boldsymbol{\theta}(t)), g(\boldsymbol{\phi}(t)))$ *converges to* $(p,q)$ *under the dynamics of Equation* (7).

*Proof.* Simple substitution gives us

$$\dot{\boldsymbol{\theta}} = -\nabla_{\boldsymbol{\theta}}f(\boldsymbol{\theta})\|\nabla_{\boldsymbol{\phi}}g(\boldsymbol{\phi})\|^2(f(\boldsymbol{\theta}) - p)$$
$$\dot{\boldsymbol{\phi}} = -\nabla_{\boldsymbol{\phi}}g(\boldsymbol{\phi})\|\nabla_{\boldsymbol{\theta}}f(\boldsymbol{\theta})\|^2(g(\boldsymbol{\phi}) - q)$$

Applying Lemma 1 and following the same steps as before

$$\dot{f} = -\|\nabla_{\boldsymbol{\theta}}f(X_{\boldsymbol{\theta}(0)}(f))\|^2\|\nabla_{\boldsymbol{\phi}}g(X_{\boldsymbol{\phi}(0)}(g))\|^2(f - p)$$
$$\dot{g} = -\|\nabla_{\boldsymbol{\phi}}g(X_{\boldsymbol{\phi}(0)}(g))\|^2\|\nabla_{\boldsymbol{\theta}}f(X_{\boldsymbol{\phi}(0)}(f))\|^2(g - q)$$

Once again we consider the function

$$H(f,g) = \int_p^f \frac{z - p}{\|\nabla f(X_{\boldsymbol{\theta}(0)}(z))\|^2}\mathrm{d}z + \int_q^g \frac{z - q}{\|\nabla g(X_{\boldsymbol{\phi}(0)}(z))\|^2}\mathrm{d}z$$

Simple substitution gives

$$\dot{H} = -(f - p)\left(\|\nabla_{\boldsymbol{\phi}}g(X_{\boldsymbol{\phi}(0)}(g))\|^2(f - p)\right) - (g - q)\left(\|\nabla_{\boldsymbol{\theta}}f(X_{\boldsymbol{\phi}(0)}(f))\|^2(g - q)\right)$$

A little bit of reorganization gives

$$\dot{H} = -(f - p)^2\|\nabla_{\boldsymbol{\phi}}g(X_{\boldsymbol{\phi}(0)}(g))\|^2 - (g - q)^2\|\nabla_{\boldsymbol{\theta}}f(X_{\boldsymbol{\theta}(0)}(f))\|^2 \leq 0$$

Thus, we get

$$\dot{H} \leq 0 \text{ in } D = \mathrm{Im}_f(\boldsymbol{\theta}(0)) \times \mathrm{Im}_g(\boldsymbol{\phi}(0))$$

Similarly with the strict convex analysis of the previous section, if $(f(\boldsymbol{\theta}(0)), g(\boldsymbol{\phi}(0))) = (p,q)$ then the theorem holds trivially. Otherwise define

$$H_0 = H(f(\boldsymbol{\theta}(0)), g(\boldsymbol{\phi}(0)))$$
$$\Omega = \{(f,g) \in D | H(f,g) \leq H_0\}$$

where we know that $H_0 > 0$ from Theorem 3. Additionally, we can apply Claim 1 even in the new dynamics, so $\Omega$ is in the interior of $D$. Since $\dot{H} \leq 0$, starting in $\Omega$, it implies that $H(f(t), g(t)) \leq H_0$ for $t \geq 0$, so $f(t), g(t)$ stays in $\Omega$. Additionally, $\Omega$ is closed since it is a sublevel set of a continuous function. Notice that the restriction of $\Omega$ on $D$ does not affect the above properties since $\Omega$ is in the interior of $D$. Thus $\Omega$ is a compact forward invariant set.

For a safe initialization $(\boldsymbol{\theta}(0),\boldsymbol{\phi}(0)$, both $\|\nabla_{\boldsymbol{\phi}}g(X_{\boldsymbol{\phi}(0)}(g(t)))\|, \|\nabla_{\boldsymbol{\theta}}f(X_{\boldsymbol{\theta}(0)}(f(t)))\|$ cannot go to 0 as this happens only at the boundaries of $D$ which are outside $\Omega$. So $\dot{H} = 0$ only at $(p,q)$ in $\Omega$.

Therefore, applying Theorem 10, we get that $(f(\boldsymbol{\theta}(t)), g(\boldsymbol{\phi}(t)))$ converges to $(p,q)$

$\square$

## C.3 Regularization and convergence

In this section, we show that even in the absence of strict convexity/concavity for both of the operators, it is possible to achieve a positive convergence result by sacrificing the exactness of a targeted equilibrium. In other words, we prove that by adding a small regularization term, the new utility function becomes strictly convex strictly concave. Beside the guaranteed convergence of the "perturbed" $L'$, we can always choose sufficiently small magnitude of regularization such that the new equilibria are arbitrarily close to the initial ones.

**Theorem 7.** *If $L$ is a convex concave function with invertible Hessians at all its equilibria, then for each $\epsilon > 0$ there is a $\lambda > 0$ such that $L'$ has equilibria that are $\epsilon$-close to the ones of $L$.*

*Proof.* For any choice of $\lambda > 0$ we have that $L'$ is strictly convex strictly concave so the KKT conditions are sufficient to determine its equilibria.

$$\frac{\partial L(\mathbf{x}, \mathbf{y})}{\partial x_i} + \lambda x_i = 0$$

$$\frac{\partial L(\mathbf{x}, \mathbf{y})}{\partial y_j} - \lambda y_j = 0$$

We can view the above set of constraints as a single vector constraint $r(\lambda, \mathbf{x}, \mathbf{y}) = \mathbf{0}$. Note that by assumption of the Hessians being invertible at all equilibria, $L$ has a unique equilibrium $(\mathbf{x}^*, \mathbf{y}^*)$. Clearly we have that $r(0, \mathbf{x}^*, \mathbf{y}^*) = \mathbf{0}$. Observe that for the Jacobian of $r$ at $(0, \mathbf{x}^*, \mathbf{y}^*)$ with respect to $(\mathbf{x}, \mathbf{y})$ we have that

$$D_{(\mathbf{x}, \mathbf{y})} r(0, \mathbf{x}^*, \mathbf{y}^*) = \nabla^2 L(\mathbf{x}^*, \mathbf{y}^*)$$

and thus it is invertible. Invoking the Implicit function Theorem, there is a differentiable function $g$, defined in a small enough neighborhood of 0, that takes a $\lambda$ and returns $g(\lambda) = (\mathbf{x}(\lambda), \mathbf{y}(\lambda))$ such that $r(\lambda, g(\lambda)) = \mathbf{0}$. Thus for a small enough $\lambda$, we have that $g$ returns the corresponding equilibria of $L'$. By continuity of $g$, for all $\epsilon$ there is a $\delta > 0$

$$\forall 0 < \lambda < \delta : \|\mathbf{x}(\lambda) - \mathbf{x}(0)\|^2 + \|\mathbf{y}(\lambda) - \mathbf{y}(0)\|^2 \le \epsilon^2$$

But $(\mathbf{x}(0), \mathbf{y}(0)) = (\mathbf{x}^*, \mathbf{y}^*)$ so the equilbrium of $L'$ has an $\epsilon$-close equilibrium of $L$ for $\lambda < \delta$. By strict convexity strict concavity of $L'$, it has a unique equilibrium as well. So the equilibria of $L'$ and $L$ are $\epsilon$-close to each other. $\square$

The previous theorem highlights that small values of $\lambda$ induce only small changes to the equilibria of the hidden game. As is the case for classical convex concave games, larger values of $\lambda$ lead to (exponentially) faster convergence. To prove this for HCC games, we provide a detailed upper and lower bound analysis of the gradients of $f_i$ and $g_j$.

**Theorem 8.** *Let $(\boldsymbol{\theta}(0), \boldsymbol{\phi}(0))$ be a safe initialization for the unique equilibrium of $L'$ $(\mathbf{p}, \mathbf{q})$. If*

$$r(t) = \|\mathbf{F}(\boldsymbol{\theta}(t)) - \mathbf{p}\|^2 + \|\mathbf{G}(\boldsymbol{\phi}(t)) - \mathbf{q}\|^2$$

*then there are initialization dependent constants $c_0, c_1 > 0$ such that $r(t) \le c_0 \exp(-\lambda c_1 t)$.*

*Proof.* Following the same analysis with the strict convex concave analysis of the previous section, if $(\mathbf{F}(\boldsymbol{\theta}(0)), \mathbf{G}(\boldsymbol{\phi}(0))) = (\mathbf{p}, \mathbf{q})$ then the theorem holds trivially. Otherwise, since our initialization is safe for $(\mathbf{p}, \mathbf{q})$, we can construct the $H$ function as in Theorem 3 and prove that it has the following property in $D = \{\text{Im}_{f_i}(\boldsymbol{\theta}_i(0))\}_{i=1}^N \times \{\text{Im}_{g_j}(\boldsymbol{\phi}_j(0))\}_{j=1}^M$

$$\dot{H} \le L'(\mathbf{p}, \mathbf{G}) - L'(\mathbf{p}, \mathbf{q}) + L'(\mathbf{p}, \mathbf{q}) - L'(\mathbf{F}, \mathbf{q})$$

$$\le -\frac{\lambda}{2} \left( \|\mathbf{F}(\boldsymbol{\theta}(t)) - \mathbf{p}\|^2 + \|\mathbf{G}(\boldsymbol{\phi}(t)) - \mathbf{q}\|^2 \right)$$

$$\le -\frac{\lambda}{2} r(t)$$

Where the second step follows from $L'(\mathbf{p}, \cdot)$ being $\lambda$ strongly concave and $L'(\cdot, \mathbf{q})$ being $\lambda$ strongly convex and $\mathbf{q}, \mathbf{p}$ being the corresponding optima of these functions since $(\mathbf{p}, \mathbf{q})$ is an equilibrium. Let us define

$$H_0 = H(\mathbf{F}(\boldsymbol{\theta}(0)), \mathbf{G}(\boldsymbol{\phi}(0)))$$
$$\Omega = \{(\mathbf{F}, \mathbf{G}) \in D | H(\mathbf{F}, \mathbf{G}) \leq H_0\}$$

where we know that $H_0 > 0$ from Theorem 3. Additionally, we can apply Claim 1 even in the new dynamics, so $\Omega$ is in the interior of $D$. Since $\dot{H} \leq 0$, starting in $\Omega$, it implies that $H(\mathbf{F}(\boldsymbol{\theta}(t)), \mathbf{G}(\boldsymbol{\phi}(t))) \leq H_0$ for $t \geq 0$, so $(\mathbf{F}(t), \mathbf{G}(t))$ stays in $\Omega$. Additionally, $\Omega$ is closed since it is a sublevel set of a continuous function. Notice that the restriction of $\Omega$ on $D$ does not affect the above properties since $\Omega$ is in the interior of $D$. Thus $\Omega$ is a compact forward invariant set.

For a safe initialization $(\boldsymbol{\theta}(0), \boldsymbol{\phi}(0))$, the following continuous functions must have a minimum and maximum value on $\Omega$ respectively.

$$K_{f_i} \geq \|\nabla f_i(X_{\boldsymbol{\theta}_i(0)}(\cdot))\|^2 \geq \kappa_{f_i}$$
$$K_{g_j} \geq \|\nabla g_j(X_{\boldsymbol{\phi}_j(0)}(\cdot))\|^2 \geq \kappa_{g_j}$$

Observe that the minima and maxima must be all greater than zero , since both $\|\nabla_{\boldsymbol{\phi}_j} g_j(X_{\boldsymbol{\phi}_j(0)}(g(t)))\|, \|\nabla_{\boldsymbol{\theta}_i} f_i(X_{\boldsymbol{\theta}_i(0)}(f(t)))\|$ cannot go to 0 as this happens only at the boundaries of $D$ which are outside $\Omega$.

Let us define

$$\kappa = \min\{\min_{1 \leq i \leq n} \kappa_{f_i}, \min_{1 \leq j \leq m} \kappa_{g_j}\}$$
$$K = \max\{\max_{1 \leq i \leq n} K_{f_i}, \max_{1 \leq j \leq m} K_{g_j}\}$$

Observe that $K \geq \|\nabla f_i(X_{\boldsymbol{\theta}_i(0)}(\cdot))\|^2 \geq \kappa$ in this interval. Thus we can write

$$\frac{(f_i(\boldsymbol{\theta}_i(t)) - p_i)^2}{2\kappa} \geq \int_{p_i}^{f_i(\boldsymbol{\theta}_i(t))} \frac{z - p_i}{\|\nabla f_i(X_{\boldsymbol{\theta}_i(0)}(z))\|^2} \mathrm{d}z \geq \frac{(f_i(\boldsymbol{\theta}_i(t)) - p_i)^2}{2K}$$

Repeating the same argument for all $f_i$ and $g_j$ we have that

$$\frac{r(t)}{2\kappa} \geq H(\mathbf{F}(\boldsymbol{\theta}(t)), \mathbf{G}(\boldsymbol{\phi}(t))) \geq \frac{r(t)}{2K}$$

Thus we can extend our analysis

$$\dot{H} \leq -\lambda r(t) \leq -\frac{2\kappa\lambda}{2} H(t) \Rightarrow H(t) \leq H_0 e^{-\lambda\kappa t} \Rightarrow r(t) \leq 2 \times K \times H_0 e^{-\lambda\kappa t}$$

$\square$

# D  Applications

## D.1  Connecting GANs and Hidden Convex-Concave Games

At the heart of many GAN formulations like the standard GAN [26], f-GAN [50] and Wassertein GAN (WGAN) [4] lies a classical convex concave game in the operator output space. Indeed for the realizeable case [26] used the underlying convexity properties to find the Nash equilibria of standard GAN and [23] did the same thing for the f-GAN and WGAN. Perhaps surprisingly, neither work references explicitly the convex concave nature of the operator output space game or von Neumann's minimax theorem. To highlight the significance of von Neumann equilibria as a solution concept for GANs, we show how the optimal $G^*$ and $D^*$ can be derived separately from each other by solving the corresponding min-max (max-min) problems. This allows one to independently verify the validity of von Neumann's minimax theorem and its generalizations for GANs. We also extend our analysis to a wide class of non-realizeable cases as well.

In practice however, as noted explicitly by [27], the updates in GAN training happen in the parameter space giving rise to a HCC game. This has exactly motivated studying the learning dynamics of HCC games in Section 3.

Thus, in this section, we present these connections between Hidden Convex-Concave games and the different architectures of Generative Adversarial Networks. More specifically, we start by exploring the structure of GANs and we verify their hidden convex-concave intrinsic form.

1. *Under this scope of hidden games*, the strong (or even strict) convexity/concavity of at least one of the players (Discriminator/Generator) in combination with the convergence results of the following sections provide some theoretical explanation about the convergence properties of those architectures even under the vanilla Gradient Descent-Ascent Dynamics.

2. To indicate the relation of Von-Neumann solution with this hidden model, we leverage this hidden convex-concave structure in order to compute the well-known both $\min\max$ and $\max\min$ optima of GANs under the realizability or not assumption. The results of this section are summarized in the following table:

| Type of GAN | $G^*$ | $D^*$ | Hidden Structure |
|---|---|---|---|
| GAN | $p_{\text{data}}$ | $\frac{1}{2}$ | Linear VS Strongly-Concave |
| GAN | $\arg\min_{G\in\mathcal{G}} \text{JSD}(p_{\text{data}}\|p_{\text{G}})$ | $\frac{p_{\text{data}}}{p_{\text{data}}+p_{\text{G}^*}}$ | Linear VS Strongly-Concave |
| f-GAN | $p_{\text{data}}$ | $f'(1)$ | Linear VS Concave |
| f-GAN | $\arg\min_{G\in\mathcal{G}} \text{D}_f(p_{\text{data}}\|p_{\text{G}})$ | $f'\left(\frac{p_{\text{data}}}{p_{\text{G}^*}}\right)$ | Linear VS Concave |
| WGAN | $p_{\text{data}}$ | $c$ | Linear VS Linear |
| WGAN | $\arg\min_{G\in\mathcal{G}} \text{EMD}(p_{\text{data}}\|p_{\text{G}})$ | – | Linear VS Linear |

Table 1: $p_{\text{data}}$ represents the target data distribution. $G^*$ is the min-max generator and $D^*$ is the max-min discriminator. JSD denotes the Jensen–Shannon divergence, $\text{D}_f$ the $f$-divergence for the convex function $f$ and EMD the earth mover distance and $c$ the constant discriminator. xGAN, xGAN correspond to the realizable and the non-realizable case accordingly. – indicates the lack of a closed form solution for $D^*$ of WGAN.

In the following three subsections, we analyze both the derivation of $\arg\min\max$ and $\arg\max\min$ for the **"vanilla-GANs", f-GANs, W-GANs** using min-max optimization arguments based on the Minimax Theorem for convex-concave functions. More precisely,

1. In the Lemmas 4, 9 and 14, we present the optimal discriminators which consist the best-response for the case of a fixed generator. In all these maximization problems, typically each $D(x)$ is decoupled and $D_G^*(x)$ is derived by the hidden concavity of the discriminator architecture.

2. In the Lemmas 5, 10 and 15, we present the optimal generators which consist the best-response for the case of a fixed discriminator. In all these minimization problems, typically the generator can cheat the fixed discriminator by producing greedily a distribution only over the restricted subset of the points for which the discriminator has the highest confidence about their originality.

3. In the Lemmas 6, 11 and 16, we leverage lemmas of (Item 1) to understand the form GAN's utility function which corresponds typically to JSD, $f$-divergence and Wasserstein distance which donate their name to their GAN architecture as well. Thus, it is then trivial to show that $p_{\text{data}}$ is the optimal choice in the realizable case.

4. In the Lemmas 7, 12 and 17, on the other side of the coin, we emphasize to derive the minmax solutions too. Our proof strategy invokes the partition to two basic sets, $S_{G_D^*}$ and $S_{G_D^*}^c$ ,the "preferable" or not data points by the generator. Leveraging the concavity part of the objective, we show that the best strategy for the discriminator is to label all the points uniformly with the same confidence in order to incentivize the generator to expands its support to the maximum possible.

5. In the Lemmas 8 and 13, we analyze the non-realizable case. One the one hand using Item 3 we are able to compute the $\arg\max\min$ generator $G^*$. To conclude about the $\arg\min\max$ discriminators we apply the Von Neumann's Minimax theorem to prove $D^* = \text{Best-Response}(G^*)$.

### D.1.1 GAN

The utility of the zero-sum game $V(G, D)$ for the distribution $p_{\text{data}}$ over the discrete set $\mathcal{N}$ is

$$V(G, D) = \sum_{x \in \mathcal{N}} p_{\text{data}}(x) \log(D(x)) + \sum_{x \in \mathcal{N}} p_{\text{G}}(x) \log(1 - D(x))$$

On the one hand, it is easy to check that for a fixed discriminator $D$, the utility function is linear over the $p_{\text{G}}$ operator. On the other hand, for a fixed generator $G$, the utility function is of the form $a \log(D) + b \log(1 - D)$ which is strongly-concave.

We start our work with the following lemmas

**Lemma 4** ([26]). *For a fixed generator $G$ the optimal discriminator is*

$$D_G^*(x) = \frac{p_{data}(x)}{p_{data}(x) + p_G(x)}$$

*Proof.* Observe that the optimization problem for each $D(x)$ is decoupled. Thus

$$D_G^*(x) = \arg\max_{D \in [0,1]} p_{\text{data}}(x) \log(D) + p_{\text{G}}(x) \log(1 - D)$$

By concavity the unique maximum of the above is given by

$$D_G^*(x) = \frac{p_{\text{data}}(x)}{p_{\text{data}}(x) + p_{\text{G}}(x)}$$

$\square$

**Lemma 5.** *For a fixed discriminator $D$, any distribution supported only on*

$$S_{G_D^*} = \{x \in \mathcal{N} : \forall x' \in \mathcal{N} \quad D(x) \geq D(x')\}$$

*is an optimal generator when it is allowed to choose any distribution over $\mathcal{N}$.*

*Proof.* Observe that for a fixed discriminator, the optimal generator optimizes

$$\sum_{x \in \mathcal{N}} p_{\mathrm{G}}(x) \log(1 - D(x))$$

since the other term is independent of the generator. Let us define the following

$$D_{\max} = \max_{x \in \mathcal{N}} D(x)$$

Then we have that

$$\sum_{x \in \mathcal{N}} p_{\mathrm{G}}(x) \log(1 - D(x)) \geq \log(1 - D_{\max})$$

with the equality being true only for distributions supported only on $S_{G_D^*}$. $\qquad \square$

**Lemma 6** ([26]). *The min-max generator is the following distribution*

$$G^* = \arg\min_{G \in \mathcal{G}} \mathrm{JSD}(p_{data} || p_G).$$

*Proof.* We can substitute in $V(G, D)$ the optimal discriminator from Lemma 4. Thus we get

$$V(G, D_G^*) = \sum_{x \in \mathcal{N}} p_{\mathrm{data}}(x) \log\left(\frac{p_{\mathrm{data}}(x)}{p_{\mathrm{data}}(x) + p_{\mathrm{G}}(x)}\right) + \sum_{x \in \mathcal{N}} p_{\mathrm{G}}(x) \log\left(\frac{p_{\mathrm{G}}(x)}{p_{\mathrm{data}}(x) + p_{\mathrm{G}}(x)}\right)$$

We can now prove that

$$V(G, D_G^*) = -\log(4) + \mathrm{KL}\left(p_{\mathrm{data}} || \frac{p_{\mathrm{G}} + p_{\mathrm{data}}}{2}\right) + \mathrm{KL}\left(p_{\mathrm{G}} || \frac{p_{\mathrm{G}} + p_{\mathrm{data}}}{2}\right)$$

$$= -\log(4) + 2\mathrm{JSD}(p_{\mathrm{data}} || p_{\mathrm{G}})$$

By minimizing $V(G, D_G^*)$, the result follows trivially. $\qquad \square$

**Lemma 7.** *The max-min discriminator is*

$$\forall x \in \mathcal{N} : D^*(x) = \frac{1}{2}$$

*when the generator is allowed choose any distribution over $\mathcal{N}$,*

*Proof.* We can substitute in $V(G, D)$ the optimal generator from Lemma 5

$$V(G_D^*, D) = \log(D_{\max}) \sum_{x \in S_{G_D^*}} p_{\mathrm{data}}(x) + \log(1 - D_{\max}) \sum_{x \in S_{G_D^*}} p_{\mathrm{G}}(x)$$

$$+ \sum_{x \notin S_{G_D^*}} p_{\mathrm{data}}(x) \log(D(x)) + \sum_{x \notin S_{G_D^*}} \underbrace{p_{\mathrm{G}}(x)}_{0} \log(1 - D(x))$$

Let us define $D_{\mathrm{small}} = \{D(x) | x \notin S_{G_D^*}\}$. Observe that if $|D_{\mathrm{small}}| > 1$ then setting $D(x) = \max(D_{\mathrm{small}})$ for each $x \notin S_{G_D^*}$ improves utility. Thus for the optimal discriminator we have $|D_{\mathrm{small}}| = 1$. Let us call $D_{\min}$ the unique element of $D_{\mathrm{small}}$. Then we have that

$$x \notin S_{G_D^*} \implies D^*(x) = D_{\min}$$

$$x \in S_{G_D^*} \implies D^*(x) = D_{\max}$$

Observe that for any combination of $D_{\max}$ and $D_{\min}$ with $D_{\max} > D_{\min}$, the constant discriminator $D_{\max}$ has higher utility. Therefore we can focus our attention on the constant discriminator $D_{\mathrm{const}}(x) = D$

$$V(G_{D_{\mathrm{const}}}^*, D_{\mathrm{const}}) = \log(D) + \log(1 - D)$$

The optimal value for $D$ is $\frac{1}{2}$ and as a result

$$D^*(x) = \frac{1}{2}.$$

$\qquad \square$

**Lemma 8** (Non-realizable case)**.** *If we assume that choice of generator $G$ is restricted in $\mathcal{G}$, a convex compact subset of the $|\mathcal{N}|$ dimensional simplex, such that $p_{data} \notin \mathcal{G}$. Then*

$$(G^*, D^*) = \left( \arg\min_{G \in \mathcal{G}} \text{JSD}(p_{data} || p_G), \frac{p_{data}}{p_{data} + p_{G^*}} \right) \text{ }5$$

*Proof.* We cannot readily apply von Neumann's minimax theorem since the $V(G, D)$ may be infinite at the boundary points of $\mathcal{D} = (0, 1)^{|\mathcal{N}|}$ for the discriminator. We can still apply Fan's Minimax Theorem

$$\min_{G \in \mathcal{G}} \sup_{D \in \mathcal{D}} V(G, D) = \sup_{D \in \mathcal{D}} \min_{G \in \mathcal{G}} V(G, D).$$

It is easy to check that Lemma 6 holds even in the non-realizable case. As a result, the generator is minimizing $\text{JSD}(p_{\text{data}} || p_{\text{G}})$ whose value is finite. Clearly the quantities above are finite. Thus there exists a real number $v$, the value of the game, such that:

$$\begin{cases} \forall D \in \mathcal{D} & : V(G^*, D) \leq v = V(G^*, D^*) \quad (A) \\ \forall G \in \mathcal{G} & : V(G, D^*) \geq v = V(G^*, D^*) \quad (B) \end{cases}$$

for $G^*$ the minimizer of $\text{JSD}(p_{\text{data}} || p_{\text{G}})$ and a $D^* \in [0, 1]^{|\mathcal{N}|}$. Now applying Lemma 4, we have that

$$\tilde{D} = \text{Best-Response}(G^*) = \frac{p_{\text{data}}}{p_{\text{data}} + p_{\text{G}^*}}.$$

Additionally, by the optimality of the response and the consequence (A) of Minimax Theorem it holds that $V(G^*, \tilde{D}) = v$. Finally, since $V(G^*, \cdot)$ is strongly concave, all other discriminators receive value less than $v$ and are not optimal. Thus

$$D^* = \tilde{D} = \frac{p_{\text{data}}}{p_{\text{data}} + p_{\text{G}^*}}$$

$\square$

### D.1.2 f-GAN

The utility of the zero-sum game $V(G, D)$ for the distribution $p_{\text{data}}$ over the discrete set $\mathcal{N}$ is

$$V(G, D) = \sum_{x \in \mathcal{N}} p_{\text{data}}(x)D(x) - \sum_{x \in \mathcal{N}} p_{\text{G}}(x)f^*(D(x))$$

We will assume that $f$ is a strictly convex function with $f(1) = 0$. On the one hand, it is easy to check that for a fixed discriminator $D$, the utility function is linear over the $p_{\text{G}}$ operator. On the other hand, for a fixed generator $G$, the utility function is of the form $aD - bf^*(D)$ which is strictly-concave.

We start our work with the following lemmas

**Lemma 9** ([50])**.** *For a fixed generator $G$ the optimal discriminator is*

$$D_G^*(x) = f' \left( \frac{p_{data}(x)}{p_G(x)} \right)$$

*Proof.* Observe that the optimization problem for each $D(x)$ is decoupled. Thus

$$D_G^*(x) = \arg\max_D p_{\text{data}}(x)D - p_{\text{G}}(x)f^*(D)$$

By concavity the unique maximum of the above is given by Fermat criterion

$$D_G^*(x) = ((f^*)')^{-1} \left( \frac{p_{\text{data}}(x)}{p_{\text{G}}(x)} \right) = f' \left( \frac{p_{\text{data}}(x)}{p_{\text{G}}(x)} \right)$$

$\square$

---

5We note that $D^*$, may take the value 1 for some $x \in \mathcal{N}$ if the generator $G^*$ does not have full support. Assigning $D(x) = 1$ for some $x$ may lead to infinite utilities in general. We prove however for that the pair $(G^*, D^*)$ this is not the case. We thus consider that pair an equilibrium.

**Lemma 10.** *For a fixed discriminator $D$, any distribution supported only on*

$$S_{G_D^*} = \{x \in \mathcal{N} : \forall x' \in \mathcal{N} \quad f^*(D(x)) \geq f^*(D(x'))\}$$

*is an optimal generator when it is allowed to choose any distribution over $\mathcal{N}$.*

*Proof.* Observe that for a fixed discriminator, the optimal generator optimizes

$$-\sum_{x \in \mathcal{N}} p_{\mathrm{G}}(x) f^*(D(x))$$

since the other term is independent of the generator. Let us define the following

$$F_{\max} = \max_{x \in \mathcal{N}} f^*(D(x))$$

Then we have that

$$-\sum_{x \in \mathcal{N}} p_{\mathrm{G}}(x) f^*(D(x)) \geq -F_{\max}$$

with the equality being true only for distributions supported only on $S_{G_D^*}$. $\square$

**Lemma 11** ([50])**.** *The min-max generator is the following distribution*

$$G^* = \arg\min_{G \in \mathcal{G}} \mathrm{D}_f(p_{data} \| p_G).$$

*Proof.* We can substitute in $V(G, D)$ the optimal discriminator from Lemma 9. Thus we get

$$V(G, D_G^*) = \sum_{x \in \mathcal{N}} p_{\mathrm{data}}(x) f'\left(\frac{p_{\mathrm{data}}(x)}{p_{\mathrm{G}}(x)}\right) - \sum_{x \in \mathcal{N}} p_{\mathrm{G}}(x) f^*\left(f'\left(\frac{p_{\mathrm{data}}(x)}{p_{\mathrm{G}}(x)}\right)\right)$$

We will first prove that:

$$V(G, D_G^*) = \mathrm{D}_f(p_{\mathrm{data}} \| p_{\mathrm{G}})$$

Let's recall firstly the definition of f-divergence:

$$\mathrm{D}_f(p_{\mathrm{data}} \| p_{\mathrm{G}}) = \sum_{x \in \mathcal{N}} p_{\mathrm{G}}(x) f\left(\frac{p_{\mathrm{data}}(x)}{p_{\mathrm{G}}(x)}\right)$$

Since $f$ is convex and lower semi-continuous, Frenchel convex duality guarantees that we can write $f$ in terms of its conjugate dual as $f(u) = \sup_{v \in \mathbb{R}} \{uv - f^*(v)\}$. Equivalently we get:

$$\mathrm{D}_f(p_{\mathrm{data}} \| p_{\mathrm{G}}) = \sum_{x \in \mathcal{N}} p_{\mathrm{G}}(x) \sup_{v \in \mathbb{R}} \left\{\left(\frac{p_{\mathrm{data}}(x)}{p_{\mathrm{G}}(x)}\right) v - f^*(v)\right\}$$

$$= \sum_{x \in \mathcal{N}} \sup_{v \in \mathbb{R}} \{p_{\mathrm{data}}(x) v - f^*(v) p_{\mathrm{G}}(x)\}$$

$$= \sum_{x \in \mathcal{N}} p_{\mathrm{data}}(x) f'\left(\frac{p_{\mathrm{data}}(x)}{p_{\mathrm{G}}(x)}\right) - \sum_{x \in \mathcal{N}} p_{\mathrm{G}}(x) f^*\left(f'\left(\frac{p_{\mathrm{data}}(x)}{p_{\mathrm{G}}(x)}\right)\right)$$

The last line follows arguments similar to Lemma 9 applied for each term. By minimizing $V(G, D_G^*)$, the result follows trivially. $\square$

**Lemma 12.** *The max-min discriminator is*

$$\forall x \in \mathcal{N} : D^*(x) = f'(1)$$

*when the generator is allowed choose any distribution over $\mathcal{N}$.*

*Proof.* We want to substitute in $V(G, D)$ the optimal generator from Lemma 5. Observe that for all $x \in S_{G_D^*}$, we may not have all $D(x)$ to be equal. Only the values of $f^*$ are guaranteed to be equal, $f^*(D(x)) = F_{\max}$. However, if there are two distinct $D$ values then we can always pick the higher one and improve utility. Thus we can focus on discriminators that are constant over $S_{G_D^*}$. Let $D_{F_{\max}}$ be the corresponding value

$$V(G_D^*, D) = D_{F_{\max}} \sum_{x \in S_i} p_{\text{data}}(x) - f^*(D_{F_{\max}}) \sum_{x \in S_i} p_{\text{G}}(x)$$
$$+ \sum_{x \notin S_{G_D^*}} p_{\text{data}}(x) D(x) - \sum_{x \notin S_{G_D^*}} \underbrace{p_{\text{G}}(x)}_{0} f^*(D(x))$$

Let us define $D_{\text{small}} = \{D(x) | x \notin S_{G_D^*}\}$. Observe that if $|D_{\text{small}}| > 1$ then setting $D(x) = \max(D_{\text{small}})$ improves utility. Thus for the optimal discriminator we have $|D_{\text{small}}| = 1$. Let us call $D_{F_{\min}}$ the unique element of $D_{\text{small}}$. So for an optimal discriminator we would have a single value $D_{F_{\min}}$ with $f^*(D_{F_{\min}}) < f^*(D_{F_{\max}})$. As a result

$$x \notin S_{G_D^*} \implies D^*(x) = D_{F_{\min}}$$
$$x \in S_{G_D^*} \implies D^*(x) = D_{F_{\max}}$$

We now have two cases. For any combination with $D_{F_{\min}} > D_{F_{\max}}$, the constant discriminator $D(x) = D_{F_{\min}}$ has higher utility. Symmetrically, for any combination with $D_{F_{\max}} > D_{F_{\min}}$, the constant discriminator $D(x) = D_{F_{\max}}$ has higher utility. Thus the optimal discriminator is constant. Plugging in the constant discriminator $D_{\text{const}}(x) = D$ we get

$$V(G_{D_{\text{const}}}^*, D_{\text{const}}) = D + f^*(D)$$

The optimal value for $D$ follwoing the approach of Lemma 9 is $f'(1)$ and as a result

$$D^*(x) = f'(1)$$

$\square$

**Lemma 13** (Non-realizable case). *Assume that $f \in C^1$ is strictly convex and $\lim_{x \to 0^+} x f(\frac{1}{x})$ exists and is finite*[6]. *If the choice of generator $G$ is restricted in $\mathcal{G}$, a convex compact subset of the $|\mathcal{N}|$ dimensional simplex, such that $p_{data} \notin \mathcal{G}$ then*

$$(G^*, D^*) = \left( \arg \min_{G \in \mathcal{G}} \mathrm{D}_f(p_{data} || p_G), f'\left( \frac{p_{data}}{p_{G^*}} \right) \right)$$

*Proof.* We cannot readily apply von Neumann's minimax theorem since the $V(G, D)$ since $\mathcal{D} = \mathbb{R}^{|\mathcal{N}|}$ is not compact for the discriminator. We can still apply Fan's Minimax Theorem

$$\min_{G \in \mathcal{G}} \sup_{D \in \mathcal{D}} V(G, D) = \sup_{D \in \mathcal{D}} \min_{G \in \mathcal{G}} V(G, D).$$

It is easy to check that Lemma 12 holds even in the non-realizable case. As a result, the generator is minimizing $\mathrm{D}_f(p_{\text{data}} || p_G)$ whose value is finite under the assumptions we made on $f$. Clearly the quantities above are finite. Thus there exists a real number $v$, the value of the game, such that:

$$\begin{cases} \forall D \in \mathcal{D} & : V(G^*, D) \le v = V(G^*, D^*) \quad (A) \\ \forall G \in \mathcal{G} & : V(G, D^*) \ge v = V(G^*, D^*) \quad (B) \end{cases}$$

for $G^*$ the minimizer of $\mathrm{D}_f(p_{\text{data}} || p_G)$ and a $D^* \in \bar{\mathbb{R}}^{|\mathcal{N}|}$. Now applying Lemma 9 we have that

$$\tilde{D} = \text{Best-Response}(G^*) = f'\left( \frac{p_{\text{data}}(x)}{p_{G^*}(x)} \right).$$

Additionally, by the optimality of the response and the consequence (A) of Minimax Theorem it holds that $V(G^*, \tilde{D}) = v$. Finally, assuming that $f$ is strictly convex we get that $V(G, \cdot)$ is strictly concave, Best-Response($G^*$) is unique and thus

$$D^* = \tilde{D} = f'\left( \frac{p_{\text{data}}(x)}{p_{G^*}(x)} \right)$$

$\square$

---

[6]This assumption guarantees that the $\mathrm{D}_f$ is always finite even if the distribution chosen by the generator is not fully supported on $\mathcal{N}$. This in turn guarantees that $D^*$ is also finite resulting in a meaningful equilibrium. Unbounded divergences like KL are known to be problematic for GANs even in practice [4].

### D.1.3 WGAN

The utility of the zero-sum game $V(G, D)$ for the distribution $p_{\text{data}}$ over the discrete metric space $(\mathcal{N}, \text{dist})$

$$V(G, D) = \mathbb{E}_{\mathbf{X} \sim p_{\text{data}}}[D(\mathbf{X})] - \mathbb{E}_{\mathbf{X} \sim p_{\text{G}}}[D(\mathbf{X})]$$
$$= \sum_{x \in \mathcal{N}} (p_{\text{data}}(x) - p_{\text{G}}(x))D(x) \text{ where } \|D\|_{\text{Lip}} \leq 1$$

On the one hand, it is easy to check that for a fixed discriminator $D$, the utility function is linear over the $p_{\text{G}}$ operator. On the other hand, for a fixed generator $G$, the utility function is linear over $D$.

We start our work with the following lemmas

**Lemma 14** ([4]). *For a fixed generator $G$ the optimal discriminator is a solution of the following linear program*

$$\text{maximize over } D(\cdot) \qquad \sum_{x \in \mathcal{N}} (p_{data}(x) - p_G(x))D(x)$$

$$\text{subject to} \qquad |D(x) - D(x')| \leq \text{dist}(x, x'), \forall x, x' \in \mathcal{N}$$

*where the optimal value of the LP is the Earth mover's distance between $p_{data}$ and $p_G$.*

*Proof.* Indeed, by definition any solution of the above LP is an optimal discriminator over a fixed generator $G$. To complete the proof of the statement, we recall that Earth Mover's distance of $(p_{\text{data}}, p_{\text{G}})$ is equal to

$$\min_{\Delta \in \text{Coupling}(p_{\text{data}}, p_{\text{G}})} \mathbb{E}_{(\mathbf{X}, \mathbf{X}') \sim \Delta}[\text{dist}(X, X')].$$

Now if we consider the dual formulation of the Wasserstein distance, then the Kantorovich duality [21, 64] implies that the above linear program consists exactly the dual linear program which computes the Earth Mover's distance. $\square$

**Lemma 15.** *For a fixed discriminator $D$, any distribution supported only on*

$$S_{G_D^*} = \{x \in \mathcal{N} : \forall x' \in \mathcal{N} \quad D(x) \geq D(x')\}$$

*is an optimal generator when it is allowed to choose any distribution over $\mathcal{N}$.*

*Proof.* Observe that for a fixed discriminator, the optimal generator optimizes

$$-\sum_{x \in \mathcal{N}} p_{\text{G}}(x)D(x)$$

since the other term is independent of the generator. Let us define the following

$$D_{\max} = \max_{x \in \mathcal{N}} D(x)$$

Then we have that

$$-\sum_{x \in \mathcal{N}} p_{\text{G}}(x)D(x) \geq -D_{\max}$$

with the equality being true only for distributions supported only on $S_{G_D^*}$. $\square$

**Lemma 16** ([4]). *The min-max generator is the following distribution*

$$G^* = \arg\min_{G \in \mathcal{G}} \text{EMD}(p_{data}, p_G).$$

*Proof.* We can substitute in $V(G, D)$ the optimal discriminator from Lemma 14. Thus we get

$$V(G, D_G^*) = \text{EMD}(p_{\text{data}}, p_{\text{G}})$$

By minimizing $V(G, D_G^*)$, the result follows trivially. $\square$

**Lemma 17.** *The max-min discriminator is*

$$\forall x \in \mathcal{N} : D^*(x) = c, \text{ Constant function}$$

*when the generator is allowed choose any distribution over $\mathcal{N}$,*

*Proof.* We can substitute in $V(G, D)$ the optimal generator from Lemma 5

$$V(G_D^*, D) = D_{\max} \sum_{x \in S_{G_D^*}} p_{\text{data}}(x) - D_{\max} \sum_{x \in S_{G_D^*}} p_{\text{G}}(x)$$

$$+ \sum_{x \notin S_{G_D^*}} p_{\text{data}}(x) D(x) - \sum_{x \notin S_{G_D^*}} \underbrace{p_{\text{G}}(x)}_{0} D(x)$$

Observe that for $x \notin S_{G_D^*}$, if $D$ takes more than two values then setting $D$ equal to the highest of the them for all $x \notin S_{G_D^*}$ improves utility. So for an optimal discriminator we would have a single value $D_{\max} > D_{\min}$. In the end we have that

$$x \notin S_{G_D^*} \implies D^*(x) = D_{\min}$$
$$x \in S_{G_D^*} \implies D^*(x) = D_{\max}$$

Observe that for any combination of $D_{\max}$ and $D_{\min}$ with $D_{\max} > D_{\min}$, the constant discriminator $D_{\max}$ has higher utility. Therefore we can focus our attention on the constant discriminator $D_{\text{const}}(x) = D$, where the optimal value is exactly zero.

$$V(G_{D_{\text{const}}}^*, D_{\text{const}}) = 0$$

Finally, it is easy to check that the choice of constant discriminator satisfies trivially the Lipschitz constraints, i.e $|D_{\text{const}}(x) - D_{\text{const}}(x')| = 0 \leq \text{dist}(x, x')$ for any metric function $\text{dist}$. $\qquad\square$

## D.2 GANs and Hidden Constrained Optimization

In the following section, we will generalize the results of Section 3.2 and Section 3.3 for the case of *vanilla* GAN of [26] whose objective is linear-strong-concave where the maximization part is constrained in the distributional simplex. More precisely,

$$\min_{\substack{p_{\text{G}}(x) \geq 0, \\ \sum_{x \in \mathcal{N}} p_{\text{G}}(x) = 1}} \max_{D \in (0,1)^{|\mathcal{N}|}} V(G, D) = \sum_{x \in \mathcal{N}} p_{\text{data}}(x) \log(D(x)) + \sum_{x \in \mathcal{N}} p_{\text{G}}(x) \log(1 - D(x))$$

At a first glance, by rewriting the equivalent Langrangian formulation of the aforementioned constrained min-max problem we can see that strong-concavity property does not hold any more. However our following theorem shows that by exploiting further the structure of [26]'s architecture a convergence result is possible.

**Theorem 16.** *Let $V(Gen_{\boldsymbol{\theta}}, Disc_{\boldsymbol{\phi}})$ be Goodfellow GAN as described in Section 4, where $G, D$ use sigmoid activations. Then for a fully mixed distribution $p_{\text{data}}$, $(\mathbf{F}(t) = Gen_{\boldsymbol{\theta}(t)}, \mathbf{G}(t) = Disc_{\boldsymbol{\phi}(t)})$ converges to $(p_{\text{data}}, \frac{1}{2}\mathbf{1}_{|\mathcal{N}|})$ as $t \to \infty$ under the dynamics of Equation (1).*

*Proof.* Let us write down our original objective

$$\min_{\substack{p_{\text{G}}(x) \geq 0, \\ \sum_{x \in \mathcal{N}} p_{\text{G}}(x) = 1}} \max_{D \in (0,1)^{|\mathcal{N}|}} V(G, D) = \sum_{x \in \mathcal{N}} p_{\text{data}}(x) \log(D(x)) + \sum_{x \in \mathcal{N}} p_{\text{G}}(x) \log(1 - D(x))$$

In order to remove the constraints from the objective above, we plan to make use of a Lagrange multiplier. We remind the reader that since both the discriminator and the generator use the sigmoid activations, we only have to capture the $\sum_{x \in \mathcal{N}} p_{\text{G}}(x) = 1$ constraint. Thus, our equivalent Langragian is:

$$\min_{\boldsymbol{\theta} \in \mathbb{R}^{|\mathcal{N}|}} \max_{\boldsymbol{\phi} \in \mathbb{R}^{|\mathcal{N}|}, \lambda \in \mathbb{R}} L(\mathbf{F}, \mathbf{G}, \lambda) = \mathbf{p}_{\text{data}}^{\top} \log(\mathbf{G}(\boldsymbol{\phi})) + \mathbf{F}(\boldsymbol{\theta})^{\top} \log(1 - \mathbf{G}(\boldsymbol{\phi})) + \lambda(\mathbf{F}(\boldsymbol{\theta})^{\top} \mathbf{1}_{|\mathcal{N}|} - 1)$$

where

$$\mathbf{F}(\boldsymbol{\theta}) = \begin{bmatrix} f_1(\theta_1) & f_2(\theta_2) & \cdots & f_{|\mathcal{N}|}(\theta_{|\mathcal{N}|}) \end{bmatrix}$$
$$\mathbf{G}(\boldsymbol{\phi}) = \begin{bmatrix} g_1(\phi_1) & g_2(\phi_2) & \cdots & g_{|\mathcal{N}|}(\phi_{|\mathcal{N}|}) \end{bmatrix}$$

and $f_i$ and $g_j$ are sigmoid functions and $\theta_i$ and $\phi_j$ are their one dimensional inputs. Let's write again the equivalent dynamics of Equation (3) for the sigmoid activations and the Langrage multiplier. Applying the same steps with Theorem 4 for sigmoids:

$$\begin{cases} \dot{f}_i &= -\ f_i^2(1-f_i)^2 \frac{\partial L}{\partial f_i}(\mathbf{F},\mathbf{G}) & \forall i \in [|\mathcal{N}|] \\[2mm] \dot{g}_j &= \ g_j^2(1-g_j)^2 \frac{\partial L}{\partial g_j}(\mathbf{F},\mathbf{G}) & \forall j \in [|\mathcal{N}|] \\[2mm] \dot{\lambda} &= \sum_{i=0}^{|\mathcal{N}|} f_i - 1 \end{cases}$$

Since all initializations are safe in this game, our "generalized" Lyapunov function:

$$H(\mathbf{F},\mathbf{G},\lambda) = \sum_{i=0}^{|\mathcal{N}|} \int_{p_{\text{data}}(x_i)}^{f_i} \frac{z - p_{\text{data}}(x_i)}{z^2(1-z)^2} \mathrm{d}z + \sum_{j=0}^{|\mathcal{N}|} \int_{1/2}^{g_j} \frac{z - 1/2}{z^2(1-z)^2} \mathrm{d}z + \frac{(\lambda - \lambda^*)^2}{2}$$

where $\lambda^*$ is the Langrange multiplier at the equilibrium of the non-hidden game and $x_i$ is the $i$-th element of $\mathcal{N}$. Applying the same steps as in Lemma 3 we get that GDA approaches the largest invariant set $E$ of points $(\mathbf{F},\mathbf{G},\lambda)$ that have the following properties

$$L(\mathbf{p}_{\text{data}},\mathbf{G},\lambda) = L\left(\mathbf{p}_{\text{data}}, \frac{1}{2}\mathbf{1}_{|\mathcal{N}|}, \lambda^*\right)$$

$$L\left(\mathbf{F}, \frac{1}{2}\mathbf{1}_{|\mathcal{N}|}, \lambda^*\right) = L\left(\mathbf{p}_{\text{data}}, \frac{1}{2}\mathbf{1}_{|\mathcal{N}|}, \lambda^*\right)$$

For the first equality, we have that the value of $\lambda$ does not affect $L$ when the generator respects the sum to one constraint. Thus

$$L(\mathbf{p}_{\text{data}},\mathbf{G},\lambda) = L(\mathbf{p}_{\text{data}},\mathbf{G},\lambda^*)$$

Then we can observe that $L(\mathbf{p}_{\text{data}},\mathbf{G},\lambda^*)$ is strictly concave in $\mathbf{G}$ and given that $\frac{1}{2}\mathbf{1}_{|\mathcal{N}|}$ is its unique minimum we have that

$$L(\mathbf{p}_{\text{data}},\mathbf{G},\lambda^*) = L\left(\mathbf{p}_{\text{data}}, \frac{1}{2}\mathbf{1}_{|\mathcal{N}|}, \lambda^*\right) \implies \mathbf{G} = \frac{1}{2}\mathbf{1}_{|\mathcal{N}|}$$

Given that $E$ is an invariant set and $\mathbf{G}$ is constant in $E$, we have that $\dot{\mathbf{G}} = 0$. In other words,

$$0 = \frac{1}{2^2}\left(1 - \frac{1}{2}\right)^2 \frac{\partial L}{\partial g_j}\left(\mathbf{F}, \frac{1}{2}\mathbf{1}_{|\mathcal{N}|}, \lambda\right) \quad \forall j \in [|\mathcal{N}|]$$

As a consequence we have that

$$\frac{\partial L}{\partial g_j}\left(\mathbf{F}, \frac{1}{2}\mathbf{1}_{|\mathcal{N}|}, \lambda\right) = 0 \implies f_j = p_{\text{data}}(x_j) \quad \forall j \in [|\mathcal{N}|]$$

Once again, given that $E$ is an invariant set and $\mathbf{F}$ is constant in $E$, we have that $\dot{\mathbf{F}} = 0$

$$0 = p_{\text{data}}(x_i)^2 \left(1 - p_{\text{data}}(x_i)\right)^2 \frac{\partial L}{\partial f_i}\left(p_{\text{data}}, \frac{1}{2}\mathbf{1}_{|\mathcal{N}|}, \lambda\right) \quad \forall i \in [|\mathcal{N}|]$$

This leads to

$$\frac{\partial L}{\partial f_i}\left(p_{\text{data}}, \frac{1}{2}\mathbf{1}_{|\mathcal{N}|}, \lambda\right) = 0 \implies \lambda = \log\left(\frac{1}{2}\right) \quad \forall i \in [|\mathcal{N}|]$$

Observe that by the optimality conditions of the non-hidden game, $\lambda^*$ needs to satisfy the same equation and thus $\lambda = \lambda^*$. Clearly we have that

$$(\mathbf{F},\mathbf{G},\lambda) \in E \implies (\mathbf{F},\mathbf{G},\lambda) = \left(p_{\text{data}}, \frac{1}{2}\mathbf{1}_{|\mathcal{N}|}, \lambda^*\right)$$

Thus the dynamics converge to the unique equilibrium of the hidden game. $\qquad\square$

### D.3 Zero-Sum Games

We close this section with an application of our regularization machinery in hidden bilinear games. Hidden bilinear zero-sum games were introduced by [65] and they are formally defined as:

**Definition 11** (Hidden Bilinear Zero-Sum Game). *In a hidden bilinear zero-sum game there are two players, each one equipped with a smooth function $\boldsymbol{F} : \mathbb{R}^n \to \mathbb{R}^N$ and $\boldsymbol{G} : \mathbb{R}^m \to \mathbb{R}^M$ and a payoff matrix $U_{N \times M}$ such that each player inputs its own decision vector $\boldsymbol{\theta} \in \mathbb{R}^n$ and $\boldsymbol{\phi} \in \mathbb{R}^m$ and is trying to maximize or minimize $r(\boldsymbol{\theta}, \boldsymbol{\phi}) = \boldsymbol{F}(\boldsymbol{\theta})^\top U \boldsymbol{G}(\boldsymbol{\phi})$ respectively.*

For the special case of hidden bilinear games, [65] proved that if the dimension of the game is greater or equal than two like (e.g. akin to Rock-Paper-Scissors) then GDA dynamics tend to "cycle" through their parameter space with an even more complex behavior than a typical periodic trajectory. Specifically, the system is formally analogous to Poincaré recurrent systems (e.g. many body problem in physics). In contrast, leveraging Theorem 7, we know that by adding a small regularization term we can "break" the cycling behavior and converge to an approximate Nash Equilibrium. We close this section by presenting a comparison between the optimization portraits of GDA dynamics with the absence or not of a regularization for the archetypical game of Rock-Paper-Scissors:

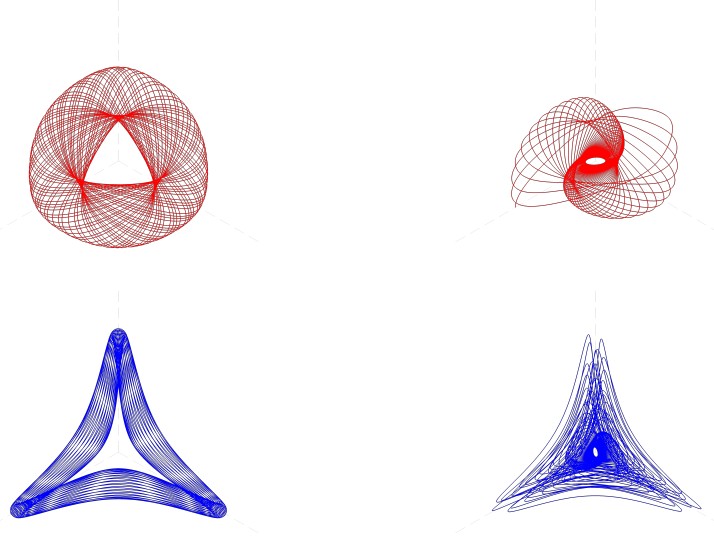

Figure 6: Trajectories of a single player using gradient-descent-ascent dynamics for a hidden bilinear game $L(\mathbf{F}(\boldsymbol{\theta}), \mathbf{G}(\boldsymbol{\phi})) = \mathbf{F}^\top(\boldsymbol{\theta}) A \mathbf{G}(\boldsymbol{\phi})$ where $A$ is the classical Rock-Paper-Scissors table and $\mathbf{F}, \mathbf{G}$ have the sigmoid activations. The two left figures present the Poincaré recurrence for different initializations of the dynamics, a behavior consistent with the Lyapunov stability of Theorem 3. On the other hand, the two figures on the right illustrate convergent to the mixed Nash equilibrium executions which exploit the regularization tools as described in Section 3.3. The regularization terms added are centered at the mixed equilbrium of the game, leading to convergence to the unmodified equilibrium of the Rock-Paper-Scissors game.

**Remark 5.** *Closing this appendix, it would be usefule to clarify some details between this work and [65]. While we use tools regarding reparametrization and safety from their work, the rest of our analysis and the technical ideas behind them are qualitatively different. [65] uses the Poincaré recurrence theorem to argue that hidden bi-linear games exhibit recurrent behavior even under safety. In contrast we show that these games are* **merely edge cases** *and that for strictly convex concave HCC convergence to the underlying Von Neumann solution is guaranteed for safe initial conditions. To the best of our knowledge, our work is the first one to provide* **sufficient conditions for non-local convergence** *to a game theoretically meaningful solution for a wide family of* **non-convex non-concave min-max** *problems. In addition, [65] require* $\mathbf{F}, \mathbf{G}$ *to be invertible operators in order to prove recurrence in the input space. In contrast, in our work we do not rely on invertibility to transfer convergence from the output space dynamics of Equation* (3) *to the input dynamics of Equation* (1).