# OpenReview forum: "Solving Min-Max Optimization with Hidden Structure via Gradient Descent Ascent"
_NeurIPS.cc/2021/Conference — NeurIPS 2021 Poster_

### Official Review · Reviewer_b7eF · 2021-07-10

**Rating:** 4
**Confidence:** 3

**Summary:**

This work studies the continuous dynamics of GDA in hidden convex-concave games via tools from dynamical systems.

**Limitations And Societal Impact:**

The author did not address the limitations and potential negative societal impact and they claim it is purely theoretical work. I would suggest adding some discussion on GAN training and biology applications.

**Main Review:**

Following ref [66] that studies hidden bilinear games, this paper studies hidden strict convex-concave games. It shows that GDA stabilizes around von Neumann solutions of hidden games. The authors claim that these hidden convex games could help us understand GAN training and this is a first step.

However, I think this paper is not ready for publication yet. During my reading I'm confused about the novelty, clarity and significance.

Clarity: I acknowledge the attempts to add some paragraphs to summarize the main contribution (line 59-63) and the structure of the paper (Line 112-115), but there are many parts of the paper that imply that this paper is not quite well written and not polished enough.

1) The question in line 38-39 comes from nowhere. Why is this question interesting? How do we relate it to previous background?
2) line 48-58: is the von Neumann solution formally defined anywhere? Because I did not find the definition.
3) line 76-78 seems repetitive
4) Line 166: what is $k$ in definition 2?
5) Thm 1: what is the meaning of $X_{\theta(0)}$, $X_{\phi(0)}$?
6) Line 233: could you explain LaSalle's invariance principle or give a reference?
7) In Section 4 it is claimed that GAN is hidden strictly convex-concave games. However, I think a caveat is that the authors are studying the game in the function space, which is convex-concave. This is already known. The nonconvexity comes from the neural net parametrization which leads to difficulty in training GANs but I have not seen it discussed anywhere.

There are many small typos which may affect the readability, such as:
1) Line 28: GANS -> GANs
2) Line 33: local minmax -> local min-max
3) Line 52: quotation marks
4) Line 151: is z(t) a definition? There is no ending mark
5) period mark in Thm 1, and the caption of Fig 3
6) The end of page 7: what is $\mathcal{N}(0, \alpha)^2$?
7) Line 281 below: what does $(0, 1)^{|\mathcal{N}|}$ mean?

Relevance & Significance:
1) I don't quite understand the biology application. What are $F, G, L$ in this application according to HCC? Also I'm not sure if this application is relevant to the ML community.
2) What is the main finding towards the practice of GANs? Is it Section 3.3 that says we should do regularization? Could you do some GAN experiments to illustrate this? How do you compare this method with say WGAN with gradient penalty?
3) GDA has been analyzed in nonconvex-concave games, see e.g.  arXiv:1906.00331. I also wouldn't say in line 365 that this paper is the first work towards understanding GAN training theoretically because there are many recent works on this, see ref [23] and refs therein.


Novelty: I would like the authors to compare their main results with ref [66]. For example,

1) Theorem 1 is similar with Lemma 2 of [66].
2)  Lemma 2 is similar to Theorem 2 of [66].

How does your proof techniques incorporate [66] and what is different?

Related work:
Line 256: Hamiltonian Gradient Descent needs some reference



**Time Spent Reviewing:**

10 hours

---

> ### Author Response · Authors · 2021-08-09
> **Response to the Official Review of Paper4987 by Reviewer b7eF**
>
>
> We thank the reviewer for their time and remarks. *Regrettably, there seems to be some misunderstandings regarding our paper’s contributions, which we hope to clarify in our point-to-point replies below.*
>
>
> # **Novelty**
> Lines 74-81 summarize our contributions compared to [66].
>
> * While the elementary principles of reparametrization are shared between the two works, as noted by the reviewer, the core of the analysis is fundamentally different leading to antithetical results.
> * While [66] treated safety as merely a best case scenario where GDA does not converge, we prove in **Theorem 2** that with mild assumptions, almost all initializations are safe and convergence is possible even with small amounts of regularization.
> * While [66] mainly proved results for **Equation 3** (similar to our **Theorem 3** and **Lemma 3**) and required invertibility to transfer results in the parameter space (like **Theorem 4**), our work studies the transfer of results between the two spaces without invertibility (see **Theorem 5, 6 and 8**). *Invertibility is a very strong assumption that trivializes the concept of hidden games as the input and operator space become equivalent under a global reparameterization (i.e. the game is not hidden).* Even trivial operators like linear functions of more than variable  $f(\theta_1, \theta_2) = \theta_1 + \theta_2$ fail to satisfy the invertibility assumption.
> * While [66] views the hidden aspect of games as an additional hurdle that GANs need to overcome to achieve convergence, our work exemplifies how regularizing GANs might be more beneficial if it happens in the operator space in order to achieve convergence (see also our comment 2 in Relevance & Significance and **Appendix C.3-D**). This is in contrast to standard optimization techniques that only consider regularization as “smoothing” the objective in the parameter space.
>
> # **Clarity Comments**
> We answer the questions in the order that reviewer asked them:
> 1. In the previous paragraph, our review of recent works in non-convex non-concave optimization showcases that in general there is no hope in identifying simple algorithms (like GDA) that converge to game theoretic solutions (like Nash equilibria). This motivates our search for a suitable subclass of non-convex non-concave problems that satisfies these criteria as explained by lines 38-39.
>
> 1. In a convex-concave game Nash Equilibria are also called von-Neumann solutions of the game referencing the minimax theorem that first proved their existence. *Thank you for catching this.*
> 1. We will compactify the lines 76-78 in one in our camera-ready version.
> 1. $k$ in **Definition 2** references the $k$ function in **Lemma 1**.
> 1. These functions are the inverses of $f_i$ and $g_j$ for the specific initialization. As discussed above **Theorem 1**, the trajectories of the system are constrained in a region where such an inverse is well defined since $f_i(t)$ uniquely identifies $\theta_i(t)$ and $g_j(t)$ uniquely identifies $\phi_j(t)$.
> 1. LaSalle’s Invariance Principle as well as a reference for it is provided in lines 658-661 of the supplementary material (**Theorem 10**).
> 1. In **Section 4** the utility functions are written in the function space but games are indeed HCCs that are played in the parameter space. For the vanilla GAN experiments, players control the parameters of sigmoid functions that output $p_{G}(x)$ and $D(x)$ respectively. Similarly for the WGAN, the generator does not control the infinite pdf values directly but only implicitly through the variance of the Gaussian.
>
> **Typos** *Thank you for catching the typos, we will correct them.*
>
> # **Relevance & Significance:**
> * The utility function $L$ is based on a two by two zero sum game with Nash equilibrium (0.75, 0.3) for the Blue/Red player respectively. There is a single $f$ and $g$ function for each player that have two inputs each:
> $f(\theta_1, \theta_2) = \operatorname{sigmoid}(\theta_1) * (1-\operatorname{sigmoid}(\theta_2)) + \operatorname{sigmoid}(\theta_2) * (1-\operatorname{sigmoid}(\theta_1))$
> with the $g$ function being similar. These functions compute an expectation of a XOR between two Bernoulli variables $B(\operatorname{sigmoid}(\theta_1))$ and $B(\operatorname{sigmoid}(\theta_2))$. This application corresponds to an evolutionary game theoretic setting and NeurIPS accepts several papers with a game theoretic concentration.
>
> * Our results indicate that taking into account the hidden nature of GANs can critically change how we view regularization. **Section 3.3** argues that regularizing the implicit hidden game may give similar guarantees to a convex concave game. Observe that the regularizers added are **non-convex non-concave in the parameter space**. This is in contrast to standard regularization approaches that try to make the objective more convex-concave. *This should not be confused with gradient penalty terms as our regularization only involves the outputs of the generator/discriminator and not the gradients of the outputs with respect to the discriminator inputs (the x_i for the Goodfellow GAN case)*.
>
> * The phrase was mainly referring to understanding the “hidden” aspects of training GANs. We will amend this to improve clarity. As the reviewer also notes, most other works do not target the special structure of the objectives in GAN training. As a result they need to make other strong assumptions to get results of a positive flavor (e.g. non-convex concave instead of non-convex non-concave as in [23]).
>
> We trust and hope that the above alleviates your concerns, and we look forward to your response and a fruitful discussion if you have any more questions about our paper.

---

### Official Review · Reviewer_Ywxk · 2021-07-14

**Rating:** 7
**Confidence:** 3

**Summary:**

In this paper, the authors study a two-player zero-sum convex-concave game whose inputs are parameterized by nonlinear functions. This results in the overall game being nonconvex-nonconcave in the parameter space. This setting is similar to that observed in GANs where the game is convex-concave in function space of discriminator and generator but is nonconvex-nonconcave in the space of parameters. The authors call such games as Hidden Convex-Concave (HCC) games.

**Limitations And Societal Impact:**

Please address the concerns mentioned in the main review.

**Main Review:**

The paper generalizes the work of Flokas et al. for hidden bilinear games. The generalization is theoretically nontrivial because Euclidean distance is not a Lyapunov function in general. The authors propose a generalized Lyapunov function for analyzing asymptotic convergence of dynamics to the solution of the HCC game. In that process, they need to assume that the underlying game is strictly convex-concave (which should not be confused with strictly-convex strictly-concave).
In the case of convex-concave games, they study a regularized underlying game. This makes the problem strictly (or rather strongly) convex-concave. While it changes the underlying equilibrium, the aforementioned results immediately imply convergence to the regularized equilibrium point.
In the case of GAN example in Section 4 (first application), the presence of \lambda (\sum_N p_G(x)-1) in the final game makes \lambda part of the maximization variable. Then, this underlying problem is not strictly concave in the combined variable (D, \lambda) for fixed G. Does this affect the convergence claims?
Overall, the authors have provided a clear explanation of their claims. The paper makes interesting contributions that might contribute to a better understanding of GANs. The point that different parameters \theta_i are used in different functions can be a source of limitation for application to the wider Neural Network-based GAN models. However, I believe the limitation outweigh the overall merits and hence I provide a high score.

**Time Spent Reviewing:**

8 hours

---

> ### Author Response · Authors · 2021-08-09
> **Response to the Official Review of Paper4987 by Reviewer Ywxk**
>
> Thank you very much for the thoughtful review, and for your positive evaluation and assessment. We reply to your precise question below:
>
> In regards to the effect of linear constraints on the strong convexity concavity of the game, **Theorem 16** in **Section D.2** in the supplementary material provides the convergence analysis for the HCC game with hidden constraints of **Figure 3**. Despite the game not being strictly convex concave in the operator space for all variables, we see that the convergence guarantees still apply.
>
> We look forward to your response and an fruitful discussion if you have any more questions about our paper and
> *Thank you again for your support!*

---

### Official Review · Reviewer_1EDT · 2021-07-16

**Rating:** 6
**Confidence:** 4

**Summary:**

This paper studies a special problem structure in min-max optimization. Specifically, the paper considers the setting when the objective function can be reparametrized into a (strict) convex-concave game, or referred to as a hidden convex-concave game in the paper.

More formally, the considered function is of the form L(F(\theta), G(\phi)), where L is convex-concave. F and G are of separable structure, namely both the max and min player are from a cartesian product of scalar functions from disjoint sets of parameters. With the separable structure, the authors can thoroughly investigate the conditions for stable GDA dynamics and the good initializations for GDA to find the equilibrium of L.

Finally, the authors give some discussion on connecting the structures to GANs and evolutionary game theory/biology.

**Limitations And Societal Impact:**

N/A

This is a theoretical work.

**Main Review:**


I believe the overall construction is pretty neat and clever. The paper is also quite nicely written and everything is easy to follow. The final discussions are quite enlightening (although not very well aligned with the main story/analysis of the paper).

I do agree this is a natural and initial step for understanding more complicated nonconvex-nonconcave objectives. However once the setting is clear, the whole analysis becomes quite obvious, and the technical part is not that challenging. It is also hard to get further inspiration on how to stabilize the training or design new optimization schemes based on the studies in this paper. Therefore my overall evaluation is a borderline accept.





on minor comments:

1. argmin of "argmax" doesn't make sense at the end of page 3.

2. some typos like 'intitializations'

**Time Spent Reviewing:**

6

---

> ### Author Response · Authors · 2021-08-09
> **Response to the Official Review of Paper4987 by Reviewer 1EDT**
>
> Firstly, we would like to thank you for your time, your remarks, and your positive assessment of our work.
>
> Below we try to address the main concerns raised with the hope that at the end of this discussion, our responses will convince you to increase your final score.
>
> Indeed, initially a reader could characterize this paper as a generalization of the already presented hidden bilinear case to the more complex hidden convex concave case. However, we would like to emphasize the novel ingredients of our work which crucially increase its applicability and importance in comparison with the prior work.
>
> ### **Comment #1**: In our work, we provide convergence results by getting rid of the invertibility* requirement for the operators.
>
> Invertibility was a very strong assumption (introduced by Flokas et al. 2019 [66]) in the bilinear case, that trivializes the concept of hidden games as the input and operator space become equivalent under a global reparameterization (i.e. the game is not hidden). For example, invertibility forces each $f_i$ and $g_j$ to have only a single parameter since they have one output and need to be smooth. Of course, [66] mainly focused on proving results of negative flavor so analyzing even simple hidden but invertible models to prove non-convergence was sufficient. Our work of course clarifies that this is merely an edge case and even small amounts of regularization can resolve the issue.
>
> By contrast, our main positive convergence results obviate the need for the invertibility assumption.  Thus, the core of our contribution is the transfer of the convergence results between the two spaces without invertibility (see *Theorem 5, 6 and 8*). The mathematical details in our theoretical analysis and our visual and verbal explanations of the leveraged concepts both in the main paper and the appendix attest to the technical challenges that our work successfully tackles.
>
> Of course, if the reviewer would like to propose simplifications in our proof scheme. We would be happy to discuss them.
>
>
> ### **Comment #2**: Our compact framework permits exploration of  multiple new directions.
>
> For example, in this stronger framework, we study the effects of regularization in the output space and its effects in the input space. Several standard tricks could be applied in the output space instead of the parameter space. Our analysis of a modified Hamiltonian Gradient Descent in *Appendix C.2.2* based on our machinery shows that our techniques are not tied to the GDA dynamics and thus many future directions may be applicable.
>
>
> Next available future steps could be :
> * The usage of high-resolution dynamical systems $[A]$ that simulates the Optimistic variant of Gradient flow in continuous HCC games.
> * The study of the  averaging in the output space as compared to averaging in the input space studied by $[B]$
> * The application of stochastic approximation framework for discretizing global asymptotically stable dynamical systems by Benaim $[C]$
>
> Notice that the broadness of all this potential future work is sparked crucially by our systematic global convergence analysis and our achievement to alleviate our model from restricting assumptions like invertibility of the operators.
>
> We trust and hope that the above alleviates your concerns, and we look forward to your response and an open-minded discussion if you have any more questions about our paper.
>
> $[A]$ http://www-stat.wharton.upenn.edu/~suw/paper/highODE.pdf
> $[B]$ https://openreview.net/pdf?id=SJgw_sRqFQ
> $[C]$ http://members.unine.ch/michel.benaim/perso/SPS99.pdf

---

### Official Review · Reviewer_Y6yb · 2021-07-16

**Rating:** 5
**Confidence:** 4

**Summary:**

This paper extends "hidden bilinear" games [1]  to "hidden convex-concave (HCC)" games, and studies a special case of HCC games (denoted as HCC* from below) where output of each dimension of a nonlinear function depends on a seperate group of parameters .

For HCC* games, the authors utilize Lyapunov-type arguments and show that GDA dynamics stabilizes around the output-space Nash equilibrium (Theorem 3), whose parameter counterparts are named as "von Neumann solutions".

With an additional assumption that the game has one-sided strictness property, the authors further show that GDA converges to a "von Neumann solution" (Theorem 6), by combining Lasalle's principle and the fact that "safe initializations" reside in ROA of such solutions.

Additionally, the authors show that adding a regularization term can accelerate the convergence of GDA towards the perturbed equilibrium in the output space (Theorem 8).

Finally, the authors exemplify GANs and evolutionary games, and argue that such games can be nicely formulated as HCC games.

**Limitations And Societal Impact:**

Please see Cons 1 & 2 for the limitations.

**Main Review:**

Pros:
1. non-convex non-concave game optimization is still an area of active research, and we still lack game classes that are (1) expressive enough to represent practical games, and (2) structured enough to admit theoretical analysis. The proposed HCC games satisfy (2).

2. Overall, the proofs seem to be clear, succinct, and well-organized. The authors also provide a brief sketch whenever the proof becomes complicated.


Cons:
1. **(Expressivity of HCC games)**  Arguably, one of the most important non-convex non-concave games in machine learning is GAN. However, it is quite dubious whether GAN can be formulated as a HCC game (even in a simplified setting). In Section 4, the authors formulate GAN training as

$\min_{p_G(x_i)} \max_{D(x_i)}  \sum_i [p_{data}(x_i) \log D(x_i) + p_G(x_i) \log(1 - D(x_i)) ]$.

However, in its original form, GAN objective follows

$\min_{\theta} \hspace{1pt} \max_{\phi} V(\theta, \phi) = \min_{\theta} \hspace{1pt} \max_{\phi} E_x[log(D_{\phi}(x))] + E_z[log(1-D_{\phi}(G_{\theta}(z)))]$,

where $\theta$ and $\phi$ represent the parameters of $G$ and $D$, respectively; the generator never learns $p_G(x_i)$ directly.

I am not sure how one could reformulate the above equation as a HCC game. Even worse, in case of HCC* games (with disjoint parameter groups assumption), I don't think the generator would be possible to model a data distribution with more than one-dimension. The significance of HCC games would be greatly improved if the authors could provide a reformulation of the above $V(\theta, \phi)$ as a HCC game $L(f(\theta), g(\phi))$ for some convex-concave $L$, smooth functions $f$ and $g$.


2. **(Novelty)**
The main idea of “hidden game” was first proposed in [1], and this paper aims to extend [1] by replacing the bilinearity with convex-concavity. To do so, the authors utilize main assumptions (“safe initialization”, the separated parameter groups), technical tools (Lemma 1, Theorem 1), and the Lyapunov function (Equation 4, invariant quantity in [1]) developed in [1]. Considering the fact that the HCC games are a generalization of hidden bilinear games, this could be a natural thing. However, it is hard to say this work is novel, since the conceptual prototype of the game class (“hidden game”) and technical tools (Lemma 1, Theorem 1, Lyapunov function) already have been presented in the preceding work.


Minor comments:
- I have an impression that Theorem 8 is quite obvious and somewhat out-of-place. Any strictly-convex strictly-concave game would admit a linear convergence for GDA, and it is unclear how it relates to the unique structure of HCC games.
- In Theorem 2, $maxLocalMin$ and $minLocalMax$ are undefined.
- In Theorem 5, $R_{f_i}$ and $R_{g_j}$ are undefined.
- Both in abstract and conclusion, the authors mention that *“our convergence guarantees are non-local, which as far as we know is a first-of-its-kind type of
result in non-convex non-concave games”*. However, I believe this is not true. For example, [2] has shown that alternating GDA globally converges for two-sided Polyak-Lojasiewicz (PL) games with a linear rate in deterministic settings, and converges with a sublinear rate in stochastic settings. This is a NeurIPS 2020 paper, so I think there would be more non-convex non-concave global convergence results published this year.

At this stage, I recommend rejection for this paper, mainly on the basis of the Cons 1 and 2.

---

[1] Poincaré Recurrence, Cycles and Spurious Equilibria in Gradient-Descent-Ascent for Non-Convex Non-Concave Zero-Sum Games. Flokas et al., NeurIPS 2019.

[2] Global Convergence and Variance Reduction for a Class of Nonconvex-Nonconcave Minimax Problems. Yang et al., NeurIPS 2020.


**Time Spent Reviewing:**

6

---

> ### Author Response · Authors · 2021-08-09
> **Response to the Official Review of Paper4987 by Reviewer Y6yb**
>
> # **Expressivity of HCC games**
>
> The equations in **Section 4** are written in the operator space representation for simplicity. In all the experiments $p_{G}(x_i)= \operatorname{sigmoid}(\theta_i)$ and $D(x_j) = \operatorname{sigmoid}(\phi_j)$ so both players learn indirectly the von Neumann solution of the operator space. Observe as well that we can have arbitrary finite support for our distribution, not merely one dimensional as the reviewer suggests, thanks to the sum to one constraints that force interactions between the $p_G$ components. The WGAN example also showcases how HCCs can learn the parameters of distributions with infinite support like Gaussians. Recapping, for finite support distributions the two objectives presented by the reviewer are equivalent and we can handle arbitrary finite support through the sum to one constraints. We can furthermore \textit{handle distributions with infinite support}, as our application on WGANs shows. In this case, the parametrization of the pdf is important. We showcase how HCCs can handle important special cases such as the WGAN setting of [36]. We hope this is clear and if not please ask follow up questions.
>
> # **Novelty**
> Lines 74-81 summarize our contributions compared to [1].
> * While the elementary principles of reparametrization are shared between the two works, as noted by the reviewer, the core of the analysis is fundamentally different leading to antithetical results.
> * While [1] treated safety as merely a best case scenario where GDA does not converge, we prove in **Theorem 2** that with mild assumptions, almost all initializations are safe and convergence is possible even with small amounts of regularization.
> * While [1] mainly proved results for **Equation 3** (similar to our **Theorem 3** and **Lemma 3**) and required invertibility to transfer results in the parameter space (like **Theorem 4**), our work studies the transfer of results between the two spaces without invertibility (see **Theorem 5, 6 and 8**). *Invertibility is a very strong assumption that trivializes the concept of hidden games as the input and operator space become equivalent under a global reparameterization (i.e. the game is not hidden).* For example, invertibility forces each $f_i$ and $g_j$ to have only a single parameter since they have one output and need to be smooth. Removing the invertibility assumption is an important technical and conceptual contribution that paves the way to more complex and realistic models that remain analytically tractable.
>
>
>
> # **Responses about the minor comments' section**
> **Theorem 8** is proving that the GDA dynamics of HCCs in the parameter space converge to the von Neumann solution $(p,q)$ not the GDA dynamics of strongly convex concave games, which of course is already known. The proof is not a straightforward application of strong Lyapunov functions as is the case for convex concave games because $r(t)$ is not monotonously decreasing. Additionally convergence rates need to be initialization dependent in contrast to the strongly convex-concave settings, as showcased by the constants $c_1$ and $c_2$. Feel free to ask more questions if this remains unclear.
>
> Regarding our claims on global convergence guarantees, we will gladly acknowledge the existing work and tone down our claim. It is however important to note that there exist trivial HCC games of one hidden dimension that are not captured in the model of [2]. Furthermore the dynamics studied are different.
>
> Regarding $R_{f_i}$ and $R_{g_j}$ not being defined in **Theorem 5**, we note that they are defined as the regular values of $f_i$ and $g_j$ and these definitions are indeed included in the theorem statement. We will include the definitions of $\textrm{LocalMin}$ and $\textrm{LocalMax}$ of **Theorem 2**.
>
> We hope that the above clarifies our choices of terminology – and we would of course be happy to explain this in the paper as well.

---

### Author Response · Authors · 2021-08-25
**Available for more discussion**

Given that the discussion period ends soon, we wanted to make sure that reviewers do not have any lingering questions. Let us know if we can help clarify something.

---

### Decision · Program_Chairs · 2021-09-27

**Decision:**

Accept (Poster)

**Comment:**

Most reviewers appreciated the novelty of the class of the problems studied and found it interesting.  There are concerns about the expressivity of HCC games in practical settings that has not been fully addressed by the authors. There were also questions about the contributions w.r.t. reference [66]. However, as authors mentioned, the invertibility assumption is one of the key differences of their work with reference [66]. Another comment was about defining various concept needed to understand the paper. It would help the reader a lot if the authors can define all necessary concepts to understand the paper (e.g. La Salle's principle Vo Neumann solution, etc.)